# Pan-Arctic seasonal cycles and long-term trends of aerosol properties from ten observatories

Julia Schmale[1], Sangeeta Sharma[2], Stefano Decesari[3], Jakob Pernov[4,1], Andreas Massling[4], Hans-Christen Hansson[5], Knut von Salzen[2], Henrik Skov[4], Elisabeth Andrews[6], Patricia K. Quinn[7], Lucia M. Upchurch[7,8], Konstantinos Eleftheriadis[9], Rita Traversi[10,11], Stefania Gilardoni[11], Mauro Mazzola[11], James Laing[12], Philip Hopke[13]

[1]Extreme Environments Research Laboratory, École Polytechnique fédérale de Lausanne, 1951 Sion, Switzerland
[2]Environment and Climate Change Canada, Science and Technology Branch, Climate Research Division, 4905 Dufferin Street, Toronto, Ontario M5T 3H4, Canada
[3]Institute of Atmospheric Sciences and Climate, National Research Council of Italy, 40129 Bologna, Italy
[4]Department of Environmental Science, iClimate, Aarhus University, Frederiksborgvej 399, 4000 Roskilde, Denmark
[5] Department of Environmental Science, Stockholm University, 106 91 Stockholm, Sweden
[6]Cooperative Institute for Research in Environmental Sciences (CIRES), University of Colorado, Boulder, Colorado 80309, USA
[7]Pacific Marine Environmental Laboratory, National Oceanic and Atmospheric Administration, Seattle, WA, USA
[8]Cooperative Institute for Climate, Ocean, and Ecosystem Studies, University of Washington, Seattle, WA, USA
[9] NCSR "Demokritos"-Institute of Nuclear and Radiological Sciences and Technology, Energy and Safety Environmental Radioactivity Laboratory, 15310 Athens, Greece
[10]Department of Chemistry "Ugo Schiff", University of Florence, 50019 Sesto F.no (Florence), Italy.
[11]Institute of Polar Sciences, National Research Council, ISP-CNR, 30172 Venice, Italy
[12] Washington State Department of Ecology, 15700 Dayton Ave N, Shoreline, WA 98133
[13] Institute for a Sustainable Environment, Clarkson University, Potsdam, NY 13699 USA

Julia Schmale ORCID 0000-0002-1048-7962
Stefano Decesari ORCID 0000-0001-6486-3786
Jakob Boyd Pernov ORCID 0000-0003-1906-2589
Rita Traversi ORCID 0000-0002-9790-2195
Patricia K. Quinn ORCID 0000-0003-0337-4895
Lucia M. Upchurch ORCID 0000-0003-1351-6332
Elisabeth Andrews ORCID 0000-0002-9394-024X
Kostas Eleftheriadis ORCID 0000-0003-2265-4905
Andreas Massling ORCID 0000-0001-8046-2798
Henrik Skov ORCID 0000-0003-1167-8696
Sangeeta Sharma ORCID 0000-0002-9071-1812

Knut von Salzen ORCID 0000-0002-2991-6181
Hans-Christen Hansson ORCID 0000-0001-7794-2889
Stefania Gilardoni ORCID 0000-0002-7312-5571
Mauro Mazzola ORCID 0000-0002-8394-2292
45       Philip Hopke ORCID 0000-0003-2367-9661

*Correspondence to*: Julia Schmale (julia.schmale@epfl.ch)

**Abstract.**

Even though the Arctic is remote, aerosol properties observed there are strongly influenced by anthropogenic emissions from outside the Arctic. This is particularly true for the so-called Arctic haze season (January through April). In summer (June through September), when atmospheric transport patterns change, and precipitation is more frequent, local Arctic sources, i.e. natural sources of aerosols
and precursors, play an important role. Over the last decades, significant reductions in anthropogenic emissions have taken place. At the same time a large body of literature shows evidence that the Arctic is undergoing fundamental environmental changes due to climate forcing, leading to enhanced emissions by natural processes that may impact aerosol properties.
In this study, we analyze nine aerosol chemical species and four particle optical properties from ten
Arctic observatories (Alert, , Kevo, Pallas, Summit, Thule, Tiksi, Barrow / Utqiagvik, Villum, Gruvebadet and Zeppelin Observatory- both at Ny-Ålesund Research Station) to understand changes in anthropogenic and natural aerosol contributions. Variables include equivalent black carbon, particulate sulfate, nitrate, ammonium, methanesulfonic acid, sodium, iron, calcium and potassium, as well as scattering and absorption coefficients, single scattering albedo and scattering Ångström exponent.
First, annual cycles are investigated, which despite anthropogenic emission reductions still show the Arctic haze phenomenon. Second, long-term trends are studied using the Mann-Kendall Theil-Sen slope method. We find in total 41 significant trends over full station records, i.e. spanning more than a decade, compared to 26 significant decadal trends. The majority of significantly declining trends is from anthropogenic tracers and occurred during the haze period, driven by emission changes between 1990
and 2000. For the summer period, no uniform picture of trends has emerged. Twenty-six percent of trends, i.e. 19 out of 73, are significant, and of those five are positive and 14 are negative. Negative trends include not only anthropogenic tracers such as equivalent black carbon at Kevo, but also natural indicators such as methanesulfonic acid and non-sea salt calcium at Alert. Positive trends are observed for sulfate at Gruvebadet.
No clear evidence of a significant change in the natural aerosol contribution can be observed yet. However, testing the sensitivity of the Mann-Kendall Theil-Sen method, we find that monotonic changes of around 5 % per year in an aerosol property are needed to detect a significant trend within one decade. This highlights that long-term efforts well beyond a decade are needed to capture smaller changes. It is particularly important to understand the ongoing natural changes in the Arctic, where
interannual variability can be high, such as with forest fire emissions and their influence on the aerosol population.
To investigate the climate-change induced influence on the aerosol population and the resulting climate feedback, long-term observations of tracers more specific to natural sources are needed, as well as of particle microphysical properties such as size distributions, which can be used to identify changes in
particle populations which are not well captured by mass-oriented methods such as bulk chemical composition.

# 1 Introduction

Despite its remoteness, the Arctic is a main receptor of anthropogenic air pollutant emissions from the northern hemisphere (AMAP, 2015, p.2015; Barrie, 1986; Quinn et al., 2007; Shaw, 1995; Heidam, 1981). Arctic air pollution has been in focus for several decades, because of the detrimental effects on ecosystems, for example through acidification (AMAP, 2006) and deposition of pollutants that bio-accumulate especially in the marine food web (Rigét et al., 2019), and due to climate impacts (Sand et

al., 2015). The Arctic Monitoring and Assessment Programme (AMAP) has issued a number of reports on the effects of short-lived climate forcers (AMAP, 2011, 2015), and found in the 2015 report that the Arctic equilibrium temperature response is in total +0.35 °C due to forcing from black carbon (BC) in the atmosphere (+ 0.40 °C), BC on snow (+0.22 °C), atmospheric organic carbon (-0.04 °C) and particulate sulfate (-0.23 °C). The summary for policy makers of the most recent AMAP report shows that the Arctic

has warmed by 0.28°C per decade between 1990 and 2015 due to reductions in $SO_2$ emissions. Reduction in BC emissions have led to a cooling of about 0.06°C per decade, whereas the $CO_2$ increase contributed 0.29°C per decade (AMAP, 2021).

Surface monitoring sites around the Arctic (Fig. 1) have been operated to determine air pollutants transported to the Arctic since as early as the 1960s at Kevo, Finland (Dutkiewicz et al., 2014; Yli-Tuomi

et al., 2003b; Laing et al., 2014a), the 1970s at Barrow / Utqiagvik, USA (Bodhaine and Dutton, 1993) and at Zeppelin, Ny-Ålesund Research Station (Platt et al., 2021), the 1980s at Alert, Canada (Sturges and Barrie, 1989), the 1990s at Villum Research Station, Greenland (Nguyen et al., 2016; Heidam et al., 1999), and more recently at a number of other observatories such as Summit, Greenland (Schmeisser et al., 2018), Tiksi, Russia (Asmi et al., 2016), Gruvebadet, Ny-Ålesund Research Station (Gilardoni, 2019;

Traversi et al., 2021), Thule, Greenland (Becagli et al., 2019) and at Pallas, Finland (Aaltonen et al., 2006; Lohila et al., 2015). Observations at these stations cover a wide range of aerosol properties including their chemical composition, i.e. particulate sulfate ($SO_4^{-2}$), nitrate ($NO_3^-$), ammonium ($NH_4^+$), mainly based on filter collection and subsequent laboratory analysis, and optical properties such as scattering and absorption coefficients and derived equivalent black carbon (EBC) concentrations. Using the Eclipse

V4.0a emission inventory and state of the art models, AMAP (2015) found that emission sources in East

and South Asia and Russia are the largest contributors to black carbon burden in the Arctic, with particularly large contributions of domestic sources in East and South Asia, and emissions from wildfires and gas flaring in Russia. Depending on the location of the observations, different source regions become relevant (Backman et al., 2021). Based on Lagrangian transport model simulations, Hirdman et al. (2010b)

showed for Zeppelin, Barrow / Utqiagvik and Alert that northern Eurasia is the main source region, whereas emissions from North America are less influential, supporting earlier work by Hopke et al. (1995), and Polissar et al. (2001). The Summit observatory located above 3000 m a.s.l. is more susceptible to emissions of Southern Europe (Hirdman et al., 2010b), since those are uplifted and arrive at the Arctic further aloft. From a near sea level surface observatory perspective, emissions in Asia and particularly

southern Asia are less important because they are transported through the higher troposphere and do not significantly influence boundary layer measurements (AMAP, 2015; Shindell et al., 2008). The seasonality of BC can be very different depending on the altitude. Mahmood et al. (2016) showed that, for example, BC in the upper troposphere is only weakly influenced by wet deposition in stratiform clouds, whereas lower tropospheric Arctic BC concentrations are highly sensitive to wet deposition.

The surface aerosol observations have created a large data resource (for a history on monitoring sites see, Platt et al., 2021), which has been analyzed for seasonal cycles. The seasonal cycle of aerosols in the Arctic is generally driven by atmospheric transport patterns, which, during winter and early spring, favor long-range transport from mid-latitudes and removal processes are minimal allowing for long aerosol lifetimes and accumulation of pollutants (e.g., Stohl, 2006). This leads to the build-up of air pollution, the

so-called Arctic haze (Heidam, 1981; Barrie, 1986; Schnell, 1984; Radke et al., 1976). During summer, the Arctic front is located far to the north and the Arctic dome largely prevents the transport of particles from sources at mid-latitude from reaching the high Arctic (Bozem et al., 2019). The summertime Arctic atmosphere is much less stratified and frequent precipitation removes aerosols efficiently, while simultaneously, conditions for long-range transport of pollution are also less frequent. Hence, the summer

boundary layer is characterized by much lower aerosol concentrations, predominantly reflecting local and regional emissions. Overall, the seasonality of aerosol properties at Arctic sites depends on the geographical sectors, and on how the specific sites are connected to upstream pollution sources through atmospheric transport. Sharma et al. (2019) analyzed annual cycles of 20 aerosol constituents at Alert. In

brief, for all anthropogenic tracers they find a winter peak and a summer minimum, as expected. For some natural tracers, such as iodine, methanesulfonic acid (MSA) and mineral dust, they find two modes, one in spring and the other in summer/fall. For mineral dust the spring peak represents long-range transported dust, while the summer/fall peak reflects more local emissions when the snow cover is at a minimum. Similarly, for MSA, an oxidation product from dimethyl sulfide (DMS), which is derived from marine microbial activity, the spring maximum reflects long-range transport, while the summer maximum is due to more local emissions. The reasons for the iodine pattern are not entirely clear but they are related likely to the return of the sun in spring and freeze-up processes in fall (Baccarini et al., 2020). For other natural tracers, such as sodium ($Na^+$) and chloride ($Cl^-$), which mainly originate from sea salt, they find a winter maximum, which is driven by sublimating blowing snow leaving behind salt particles (Huang and Jaeglé, 2017; Fenger et al., 2013). Quinn et al. (2007) reported annual cycles of anthropogenic haze tracers for Barrow / Utqiagvik, Alert, Villum, Zeppelin and several other Eurasian stations. Eckhardt et al. (2015) compiled annual BC and $SO_4^{2-}$ data from the same four stations, as well as Tiksi and Pallas in addition for a model evaluation. Backman et al. (2017) systematically evaluated aethalometer measurements of black carbon from Barrow / Utqiagvik, Alert, Zeppelin, Summit, Pallas and Tiksi, and Schmeisser et al. (2018) used the same data for a systematic analysis of seasonal cycles of aerosol optical parameters. Moschos et al. (2022a, b) used filter based analysis of major ions, black carbon and organic carbon to conduct a pan-Arctic source apportionment of total suspended particle mass and $PM_{10}$ (particulate matter with a diameter smaller than or equal to 10 µm), revealing that organic aerosol is as abundant in summer as in winter, where natural sources dominate the summer. While the seasonal patterns of the aerosol scattering and absorption coefficients follow the pattern of anthropogenic tracers in the Arctic for four of the sites, Summit and Pallas show unique signals, where the maximum is in summer rather than winter. This difference is attributed to the higher load of natural aerosol emissions from the boreal forest around Pallas (Tunved et al., 2006, 2003) and the long-range transport of fire emissions to Summit during summer (Stohl et al., 2006).

The same data as discussed above in the context of annual cycle studies have been used for the purpose of long-term trend analyses. The longest time series of black carbon measurements was collected at Kevo, Finland, where annual concentrations decreased by 73% between about 1970 and 2010 (Dutkiewicz et al.,

2014). Sharma et al. (2006) found a 33 % decline between 1989 and 2003 for Barrow / Utqiagvik during the haze season. The analysis was updated by Sharma et al. (2013), who found a roughly 40 % decrease for Barrow / Utqiagvik, Alert and Zeppelin between 1990 and 2009. More recent work by Sharma et al. (2019) extended the analysis until 2013, reporting a roughly 50 % decrease of black carbon since 1990 during the haze season. The authors ascribe the decline firstly to the economic changes in the former Soviet Union and secondly to air quality policies in Europe. The aerosol absorption coefficient can also be used to describe the influence of black carbon, an absorbing aerosol. Collaud Coen et al. (2020b) performed trend analyses and found significant trends at Barrow / Utqiagvik over a 10 year horizon of -9.91 % per year (longer periods did not have a significant trend), and -0.90 % per year at Tiksi. Alert features a positive trend of 2.77 % per year over 10 year horizons. The datasets they used ended in 2016-2018 depending on site. Observations for $SO_4^{2-}$ are analogous for Alert, also with a 50 % decline over the same period (Sharma et al., 2019). Hirdman et al. (2010a) derived trends based on annual geometric mean data and found a 21.5 % decline at Zeppelin between 1990 and 2008 in non-sea salt sulfate ($nssSO_4^{2-}$) and no trend for Barrow / Utqiagvik between 1997 and 2006. The absence of a trend at the latter station is potentially due to the short time period investigated. For Zeppelin, Platt et al. (2021) found a decline of 44 % between 1990 and 2019. The difference to findings by Hirdman et al. (2010a) can be explained by the use of different data sets and seasonal vs annual aggregates. EBC and $nssSO_4^{-2}$ are typical anthropogenic combustion tracers during the haze season, and therefore show significant negative trends at several stations during winter in the past decades (Dutkiewicz et al., 2014; Sharma et al., 2019). For summer, a much smaller number of significantly decreasing concentrations has been reported. This is partly because during summer, anthropogenic emissions are transported much less effectively to the Arctic, and EBC as well as $nssSO_4^{2-}$ can also be emitted or derived from natural sources such as wildfires and marine organisms, respectively.

The Arctic is undergoing rapid and unprecedented socio-economic as well as environmental changes. The region is warming two to three times faster than the global average, a phenomenon called Arctic amplification (Serreze and Barry, 2011). Sea ice has declined by more than 30 % since the 1970s (Meier et al., 2014), marine microbial activity is changing significantly (Arrigo et al., 2012) and up to 50% increase in wild fire occurrence in the circumboreal region is expected by the end of the century

(Flannigan et al., 2009). These changes may have important implications for natural aerosol emissions and abundance (Schmale et al., 2021), particularly in summer, when e.g., BC and OC are emitted from wild fires. Despite the increase in fire emissions in higher latitudes in the past years, no increase in the trends during the fire seasons has been shown so far.. This could be related to the wild fire plume injections heights, which might reach higher altitudes and BC and OC could then be transported further aloft and not captured by surface observations. However, knowledge on plume injections heights are still uncertain (Rémy et al., 2017; Ke et al., 2021). From carbon source apportionment studies based on isotopic ratios, it is known that biomass burning dominates the summertime EBC signal (Winiger et al., 2019). However, there are no long-term isotopic observations, and thus, it is unknown whether the summertime dominance is a new phenomenon or not. In terms of socio-economic changes, impact of emissions from oil fields (e.g., Prudhoe Bay in Alaska), shipping (e.g., in Resolute Bay, Canada), flaring (e.g., in Siberia) and Arctic urban areas have been found to change aerosol composition as well as cloud properties (Kolesar et al., 2017; Aliabadi et al., 2015; Schmale et al., 2018; Gunsch et al., 2017).

$NssSO_4^{2-}$ can originate from the oxidation of DMS to $SO_2$ and further to sulfuric acid (Hoffmann et al., 2016), which can then condense onto preexisting particles and contribute to cloud condensation nuclei or form new ones (Hodshire et al., 2019; Beck et al., 2020; Schmale and Baccarini, 2021; Park et al., 2021). DMS is also converted into MSA in the atmosphere (Hoffmann et al., 2016). It is hypothesized that retreating sea ice allows for more algal blooms in the Arctic Ocean, which would result in more DMS emissions and hence more particulate MSA, a unique tracer for this change. Gali et al. (2019) showed a significant increase in DMS emissions between 1998 and 2016 north of 70°N. A few studies have investigated the trends of particulate MSA and summertime $nssSO_4^{-2}$ in the Arctic. Recently, Moffett et al. (2020) found for Barrow / Utqiagvik that MSA and $nssSO_4^{-2}$ increased by 2.5 % and 2.1 % per year (July to September), respectively, in the period 1998 to 2017. The authors find a positive correlation with ambient temperature and hypothesize that the trend might result from a mixture of causes, which include retreating sea ice, warmer sea surface temperature and temperature dependent atmospheric chemical reactions. At Alert, Sharma et al. (2019) found no significant trend in MSA between 1998 and 2013. However, there was a 4 % increase per year (July to August) between 1998 and 2008 (Sharma et al.,

2012), reflecting natural variability rather than the effect of sea ice retreat. MSA measured at Kevo showed an increasing monthly trends in June and July from 1980 to 2010 for June. In July, the trend peaked in 2002-2003 and then declined (Laing et al., 2013). Overall they found a strong correlation of MSA in June and July with the monthly sea surface temperature anomalies from 30°W to 80°E and 40°N to 90°N. A study of MSA concentrations at Gruvebadet, Svalbard, and Thule, Greenland between 2010 and 2017 showed that the relationship between retreating sea ice and particulate MSA is complex and likely not direct (Becagli et al., 2019). While concentrations seem to have a positive trend at Gruvebadet, they tend to decline at Thule, while sea ice is retreating in both locations. The presence of algae responsible for the production of DMS precursors in the water is necessary for DMS release to the atmosphere and subsequent production of MSA. A modeling study has shown that even if more DMS is emitted, it does not mean that higher particulate MSA is observed at the surface stations, because MSA might be incorporated into cloud condensation nuclei (CCN) which are preferentially removed from the atmosphere through wet deposition further amplified by the fact that there is an increase in precipitating clouds with retreating sea ice (Mahmood et al., 2019).

Another important natural aerosol component around the Arctic is sea spray, composed of sea salt and organics. Modeling studies have investigated the climatic effects following the hypothesis that retreating sea ice will lead to more sea spray production (Browse et al., 2014; Struthers et al., 2011). However, long-term observations that show an increased contribution of sea spray are scarce. Increasing trends in sea salt produced by the sea spray were identified at Alert in 34 years of measurements of $Na^+$ and $Cl^-$ (Sharma et al., 2019). Recently, Heslin-Rees et al. (2020) have analyzed long-term trends of aerosol optical properties from Zeppelin, Svalbard. They found that the scattering Ångström exponent (*SAE*) became significantly smaller over time. A smaller SAE indicates in general larger particle diameters, and hence the results provide evidence for a change in particle diameters towards larger sizes. Given that coarse mode particle sources in the Arctic are limited, long-range transported mineral dust or local soil dust are other potential sources, but sea spray is the main contributing factor. The authors find, however, that the increased coarse mode is a result of changing atmospheric air mass advection patterns rather than enhancement of sea spray production in regions with retreated sea ice. In winter, $Na^+$ is dominated by blowing snow signals. Higher sea salt contributions at Alert from January to March (Sharma et al., 2019)

are found to be due to wind driven resuspension of sea salt originally deposited on the snowpack from long range transport of sea salt from the North Atlantic Ocean. This contribution is more important than that from frost flowers (Huang and Jaeglé, 2017) as supported by sulfur isotope analysis (Seguin et al., 2014). May et al. (2016)investigated particulate $Na^+$ at Barrow / Utqiagvik in winter and found that sea spray emissions most likely from open leads or open water were responsible for a considerable fraction of sea salt, rather than blowing snow. This might be an indication for a shift in sea salt sources during winter. However, no long-term analyses have been done. Generally, the Arctic is windier in winter than in summer, with the highest wind speeds between January and April, and the lowest in July and August, based on the 40 year ERA5 climatology (Rinke et al., 2021). Based on a trend analysis on several re-analysis products Vessey et al. (2020) found that there is no significant change in storminess in any season in the Arctic. This does however not mean that there are not Arctic regional changes over time, just that they are not captured by Arctic wide data aggregates (Atkinson, 2005).

Summarizing the large body of literature treating annual cycles and long-term trends at Arctic observatories, we find three main approaches: (i) studies investigated a limited number of aerosol properties across multiple stations for a short period (e.g. the systematic comparison of optical properties (Schmeisser et al., 2018) or BC and $SO_4^{2-}$ (Eckhardt et al., 2015), (ii) studies focused on one station entirely reporting a large number of parameters for seasonal cycle and trend analyses (e.g., Sharma et al., 2019), or (iii) studies discussed long-term trends of a small number of anthropogenic tracers across multiple stations over different periods (e.g., Quinn et al., 2007; Collaud Coen et al., 2020b). The resulting gaps are a missing pan-Arctic synthesis of anthropogenic and natural seasonal and long-term trend patterns over comparable periods, as well as a joint consideration of chemical and optical aerosol properties and their seasonal and long-term co-evolution.

To close these gaps, here, we discuss seasonal cycles of nine chemical and four optical properties across ten stations averaged over the same time-period, and we derive decadal trends per variable and station, which can be compared to each other. We further include anthropogenic as well as natural aerosol tracers.

The purpose of this study is twofold: (i) we aim to reveal similarities and differences in seasonal patterns to highlight the regional nature of the Arctic aerosol regimes, and (ii) to contrast anthropogenic and natural aerosol long-term trends against decreasing anthropogenic emissions and discuss these trends in the

context of environmental changes. This study follows chapter 5 of the recently released assessment report of short-lived climate forcers in the Arctic by the Arctic Monitoring and Assessment Programme (AMAP, 2021).

## 2 Methods

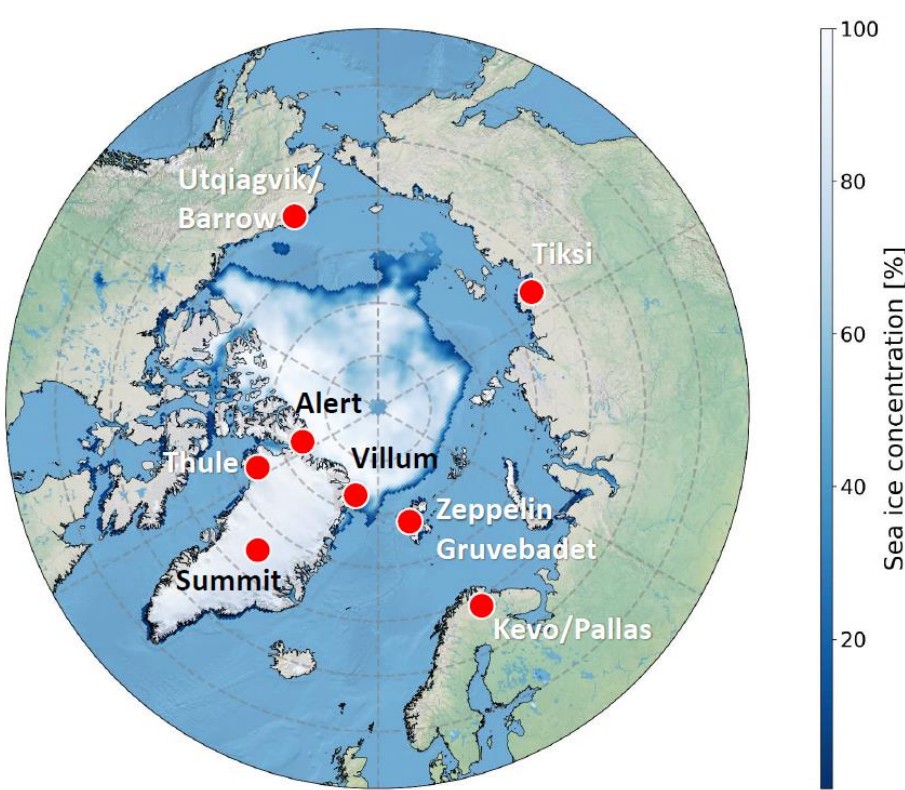

Figure 1: Locations of observatories discussed in this study. The blue shading represents sea ice concentration (15 September 2018), reported as a fraction of covered surface (Maslanik and Stroeve, 1999). The base map is from Natural Earth II (https://www.naturalearthdata.com/).

## 2.1 Data and data treatment

We obtained final, corrected data either from a publicly accessible data repository (EBAS, http://ebas.nilu.no/) or through personal communication with station PIs (see Table 1). Data temporal resolution varied from hourly for aerosol optical properties to several days for chemical components. Figure 1 shows the location of the ten observatories where various in situ measurements were conducted. The chemical variables we consider include particulate sulfate ($SO_4^{2-}$), nitrate ($NO_3^-$), ammonium ($NH_4^+$),

MSA, sodium ($Na^+$), non-sea salt calcium ($nssCa^{2+}$), non-sea salt potassium ($nssK^+$) and iron (Fe). We calculated the nss components $Ca^{2+}$ and $K^+$ based on their abundance in sea salt relative to the $Na^+$ concentration (Pilson, 2012). The former can be regarded as tracer for soil dust and the latter for biomass burning (e.g., Sharma et al., 2019). Table 1 gives an overview of the measurement methodology for each species at the different sampling sites. We also consider equivalent black carbon (EBC), which was

derived either from filter absorption measurements, including aethalometers. The details of conversion from attenuation and absorption measurements are provided in Table 2.

We also report four different optical properties, the scattering ($\sigma_{sp}$) and absorption ($\sigma_{ap}$) coefficients at 550 nm, single scattering albedo (*SSA*) at 550 nm, and scattering Ångström exponent at 550/450 (*SAE*). While the first two variables depend on the total aerosol concentration, i.e. extensive properties, the *SSA*

and *SAE* are intensive properties which do not depend on loading. The *SSA* is a measure of how much the aerosol population is able to scatter light relative to the extinction (absorption+scattering), i.e. a value of *SSA* equal to 1 indicates a purely scattering aerosol. *SSA* was calculated by dividing the scattering coefficient by the sum of the scattering and absorption coefficients. The *SAE* describes the wavelength dependence of the aerosol scattering coefficient and can be interpreted as a measure of particle size, where

larger particles have smaller *SAE* values (Delene and Ogren, 2002). We calculated *SAE* based on the following Eq. 1:

$$SAE = -\frac{\log(\sigma sp_{450}) - \log(\sigma sp_{550})}{\log(\lambda_{450}) - \log(\lambda_{550})} \qquad\qquad \text{Equation 1}$$

where $\sigma_{sp}$ is the scattering coefficient at the indicated wavelength ($\lambda$). Values below the limit of detection of 0.045 $Mm^{-1}$ for $\sigma_{sp}$ were not used in the calculation of the *SAE* since they were found to be close to

detection limit. Table 2 provides details for the conversion to $\sigma_{ap}$ at 550 nm from other wavelengths.

Not all species are available for every station. Table 1 gives an overview of the variables and the periods for which they were obtained at each station.

In this study, we only used quality checked and assured datasets from the sources provided in Table 1. EBAS data are level 2, and only data points with a red flag code as given in the EBAS data submission 325 manual (https://ebas-submit.nilu.no/temps) were omitted. Data from other sources or personal communication are equivalent to EBAS standards.

**Table 1: Overview over methods applied to derive aerosol properties and the respective data**
**sources. If a manuscript is given as data source, data were obtained in their published form from the authors.**

| Station | Variables and measurement period | Method | Remark | Data Source |
|---|---|---|---|---|
| Alert | $SO_4^{2-}$, $NH_4^+$, $Na^+$, $K^+$ (1980 – 2019) $NO_3^-$, MSA (1981 – 2019) | HI-volume sampler, ion chromatography, | Total suspended particulates (TSP) | EBAS* (Sharma et al., 2019) |
| | EBC (1990 – 2017) | Aethalometer AE06 | 880 nm, TSP | EBAS (Sharma et al., 2019) |
| | $Ca^{2+}$ (1980 – 2012), Fe (1980 – 2009) | Instrumental neutron activation analysis (INAA), inductively coupled plasma mass spectrometry (ICP-MS) | TSP INAA: 1988 – 2012 for Ca ICP: 1980 – 1997 for Ca, 1980 – 2009 for Fe, data were averaged when both methods are available | (Sharma et al., 2019) |

| | Scattering coefficient (2005 – 2018) | Nephelometer (TSI 3563) | 550 nm, particulate matter < 1 µm ($PM_1$) | S. Sharma, personal communication |
|---|---|---|---|---|
| | Absorption coefficient (2005 – 2018) | Particle soot absorption photometer, continuous light absorption photometer (PSAP/CLAP) 1-λ and 3-λ | $PM_1$ | S. Sharma, personal communication |
| Barrow / Utqiagvik | $SO_4^{2-}$, $NO_3^-$, $NH_4^+$, MSA, $Na^+$, $K^+$ (1998 – 2014) | Ion chromatography, $PM_1$ (supermicron data are also available but were not used here) | | (Quinn et al., 2007), submicron: https://data.pmel.noaa.gov/pmel/erddap/tabledap/submicron.html |
| | EBC (1992 – 2019) | Aethalometer and PSAP | $PM_{10,}$ measured or converted to 880 nm, AE8 until 2001, PSAP 1λ 2001 – 2006, PSAP 3λ 2006 – 2010, AE31 from 2010 – 2016, AE33 after 2016 | (Sharma et al., 2013), and for newer values personal communication E. Andrews |
| | Scattering coefficient (1978 – 2020) | Nephelometer (TSI 3563) | 550 nm, $PM_{10}$ | EBAS |

| | Absorption coefficient (1997 – 2019) | PSAP, CLAP | $PM_{10}$, 1$\lambda$ PSAP 1997-2006 3$\lambda$ PSAP 2006 – 2014 3$\lambda$ CLAP 2014 - present | EBAS |
|---|---|---|---|---|
| Zeppelin | $SO_4^{2-}$, $Na^+$, $Ca^{2+}$ (1993 – 2020) MSA (1990 – 2004) $NH_4^+$, $K^+$ (2010 – 2020) | Ion chromatography | $PM_{10}$ | EBAS |
| | EBC (2001 – 2017) | Aethalometer (AE31) | 880 nm, TSP | personal communication: K. Eleftheriadis, and (Eleftheriadis et al., 2009) |
| | Scattering coefficient (1999 – 2017) | Nephelometer (TSI 3563) | TSP | (Heslin-Rees et al., 2020) |
| | Absorption coefficient (2001 – 2017) | PSAP 1-$\lambda$ | $PM_{10}$ | EBAS |
| Villum | $SO_4^{2-}$, $NO_3^-$, $NH_4^+$, $Na^+$, $K^+$, $Ca^{2+}$, Fe (1991 – 2002; 2007 – 2017) | Ion chromatography, proton induced X-ray emission (PIXE) , ICP-MS | $SO_4^{2-}$, $NO_3^-$, $NH_4^+$, $Na^+$: Ion chromatography PIXE: 1991 – 2009 for $K^+$, $Ca^{2+}$, Fe ICP-MS: 2011 – 2017 for $K^+$, $Ca^{2+}$, Fe TSP | A. Massling, personal communication |

| | | | | |
|---|---|---|---|---|
| Gruvebadet | $SO_4^{2-}$, $NO_3^-$, MSA (2010 – 2019) | Ion chromatography, | $PM_{10}$ | R. Traversi, personal communication (Becagli et al., 2019, 2016) |
| | EBC (2010 – 2020) | PSAP 3-$\lambda$ | TSP | (Gilardoni, 2019) |
| | Absorption coefficient (2010 – 2020) | PSAP 3-$\lambda$ | TSP | (Gilardoni, 2019) |
| | Scattering coefficient (2010 – 2020) | Nephelometer 1-$\lambda$ (530 nm) | TSP | Personal communication: S. Gilardoni |
| Kevo | EBC (1965 – 2010) | Thermal optical methods | TSP | (Dutkiewicz et al., 2014) |
| | $Na^+$, $K^+$, $Ca^{2+}$, MSA, $SO_4^{2-}$ (1964 – 2010) | Ion chromatography | TSP | this study (Laing et al., 2013) |
| Tiksi | EBC (2009 – 2018) | Aethalometer (AE31) | 880 nm, $PM_{10}$ | EBAS |
| | Scattering coefficient (2015 – 2020) | Nephelometer (TSI 3563) | 550 nm, not controlled humidity, but heated inlet, $PM_{10}$ Note, measurements at ambient humidity > 40 % might result in larger values compared to RH< 40 % values. | EBAS |

| | Absorption coefficient (2009 – 2018) | Aethalometer (AE31) | $PM_{10}$ | EBAS |
|---|---|---|---|---|
| Thule | $SO_4^{2-}$, $NO_3^-$, MSA (2010 – 2020) | Ion chromatography | $PM_{10}$ | (Becagli et al., 2019) |
| Summit | Scattering coefficient (2012 – 2020) | Nephelometer (TSI 3563) | 550 nm, $PM_{2.5}$ | EBAS |
| | Absorption coefficient (2011 – 2020) | CLAP 3-λ | $PM_{2.5}$ | EBAS |
| Pallas | $SO_4^{2-}$, $NH_4^+$, $Na^+$, $Ca^{2+,}K^+$ (2002 – 2020), $NO_3^-$ (2013 – 2020) | Ion chromatography | TSP | EBAS |
| | Scattering coefficient (2000-2020) | Nephelometer (TSI 3563) | TSP until 2008, from 2008 $PM_{10}$ | EBAS |
| | Absorption coefficient (2007 - 2020) | MAAP (Thermo_5012) | $PM_{10}$ | EBAS |

* http://ebas-data.nilu.no/

**Table 2: Conversion factors for absorption coefficients at 550 nm and for the derivation of EBC concentrations.**

| **Station** | **Absorption coefficient** | **Instrument used to derive EBC** | **MAC** | **$C_{ref}$ value** | **References for conversions** |
|---|---|---|---|---|---|
| | | | | | |

| | **conversion to 550 nm** | | | | |
|---|---|---|---|---|---|
| **Alert** | PSAP / CLAP: λ = 530 nm, converted to $\sigma_{ap}$ at 550 nm assuming an Ångström exponent of 1 | Aethalometer, AE06, 880 nm | 19 m$^2$/g | no $C_{ref}$ applied | (Sharma et al., 2004, 2006, 2013, 2019) |
| | | Aethalometer, AE31 880 nm | 16.6 m$^2$/g | no $C_{ref}$ applied | (Sharma et al., 2004, 2006, 2013, 2019) |
| **Barrow / Utqiagvik** | PSAP: λ = 528 nm converted to $\sigma_{ap}$ at 550 nm assuming an Ångström exponent of 1 | PSAP 1-λ | 10 m$^2$/g | | (Bond et al., 1999) |
| | PSAP: λ = 530 nm converted to $\sigma_{ap}$ at 550 nm assuming an Ångström exponent of 1 | PSAP 3-λ | 10 m$^2$/g | | (Bond et al., 1999; Ogren, 2010) |
| | CLAP: λ = 528 nm converted to $\sigma_{ap}$ at 550 nm assuming an Ångström exponent of 1 | CLAP 3-λ | 10 m$^2$/g | | (Bond et al., 1999; Ogren, 2010) |
| **Zeppelin** | | Aethalometer, AE 31, 880 nm | 15.9 m$^2$/g | no $C_{ref}$ applied | (Eleftheriadis et al., 2009) |

| | | | | | |
|---|---|---|---|---|---|
| | PSAP $\lambda$ = 528 nm converted to $\sigma_{ap}$ at 550 nm assuming an Ångström exponent of 1 | | | | (Bond et al., 1999; Ogren, 2010) |
| **Gruvebadet** | PSAP $\lambda$ = 530 nm converted to $\sigma_{ap}$ at 550 nm assuming an Ångström exponent of 1 | PSAP 3-$\lambda$ | 10 m$^2$/g | | This study |
| **Tiksi** | Aethalometer AE31, $\lambda$ = 880 nm, MAC = 16.6 m$^2$/g, $C_{ref}$ = 3.5 for conversion to $\sigma_{ap}$ at 550 | Aethalometer AE 31, 880 nm | 16.6 m$^2$/g | no $C_{ref}$ applied | (Popovicheva et al., 2019) |
| **Summit** | CLAP, $\lambda$ =550 nm | CLAP 3-$\lambda$ | 10 m$^2$/g | | (Bond et al., 1999; Ogren, 2010) |
| **Pallas** | MAAP, $\lambda$ = 637 nm, converted to $\sigma_{ap}$ at 550 nm assuming an Ångström exponent of 1 | | | | This study |

## 2.2 Annual cycles and trend calculations

For the annual cycles, we calculated the median and interquartile range from all data points per month for
the available periods. To derive the anomaly, we subtracted the overall median value and normalized by

the overall interquartile range. The overall values are based on all data entries available for the given time period. They are shown in the box plots in Figure 2. Note that data for Gruvebadet are available only since 2018 for all winter months.

We derived long-term trends for all variables, where data from at least five consecutive years were available. Trends are calculated for the haze season (January, February, March, April, JFMA) and the summer season (June, July, August, September, JJAS) following the example of Quinn et al. (2009) for data obtained at Barrow / Utqiagvik. This allows for comparison with previously derived trends for aerosol properties at Arctic stations. The haze season is dominated by anthropogenic emissions, while the summer season reflects a mix of anthropogenic and natural emissions. The seasonal separation hence allows tracking the evolution of anthropogenic and natural influences more specifically. It also avoids the need for prior deseasonalization, i.e. removal of the seasonal component, of the data. Note, for MSA, we divided the seasons into April-May, and June-August, because there were not enough data above the detection limit before and after. A similar approach to separate data to derive seasonal trends has been introduced by Hirsch et al. (1982). Trend significance was tested with the non-parametric Mann-Kendall test on each season. Non-parametric means that no prior assumption of a specific distribution is necessary. The test has been widely applied to environmental data sets including in the Arctic (Tunved and Ström, 2019; Skov et al., 2020; Heslin-Rees et al., 2020; Collaud Coen et al., 2020b, a), not least because of its non-parametric nature and its tolerance of missing values. The null hypothesis of the test is that no trend exists. We defined a trend to be significant if the p-value is smaller or equal to 0.05 (green dots and upward pointing arrows in Figure 3). Trends with a p-value between 0.05 and 0.1 are marked orange and larger values (p-value > 0.1) in red (Figure 3). The results of the Mann-Kendall test are strongly dependent on autocorrelation, whereby a positive autocorrelation increases the likelihood that a trend is determined when there is none (Kulkarni and von Storch, 1995). To avoid an autocorrelation signal, and to accommodate the highly diverse time resolution of the data set, we calculate seasonal median values. The slopes of the trends were calculated using the Theil-Sen slope estimator, another non-parametric method. Significance and slope were derived for entire time series and on a decadal basis. Values for specific decades allow for intercomparison between variables and stations. However, decadal values are also given if there are at least five consecutive years of data within a decade. Hence the time periods within decades

are not equal in all cases and for all stations. Figure 3 shows with the grey bars for which years which variables are available per station. At Villum, a large time gap exists between 2003 and 2007. We derived trends separately for before and after the gap.

It is important to note that the Mann-Kendall test requires a monotonic trend for the time period over which it is calculated. This is often the case for anthropogenic tracers during the haze season. However, some variables do not show a monotonic behavior. For example, at Alert MSA exhibited a significant negative trend between 1981 and 1999 and a non-significant positive trend thereafter (Sharma et al., 2019). To reflect this, we split the data set and calculated trends separately.

## 3 Results

## 3.1 Annual cycles

### 3.1.1 Chemical composition

Figure 2 shows the statistical distribution of concentrations per chemical species, as box plots for each station on the right side and as monthly anomalies of the normalized concentrations as annual cycles on the left side.

$SO_4^{2-}$ makes up the highest concentrations at each station with medians between 0.1 and 0.3 $\mu g$ $m^{-3}$, followed by $Na^+$ (0.05 to 0.15 $\mu g$ $m^{-3}$). $NH_4^+$, $NO_3^-$, EBC, Fe and $nssCa^{2+}$ all range between 0.01 and 0.06 $\mu g$ $m^{-3}$. $NssK^+$ (except for Kevo) and MSA exhibit smaller concentrations between 0.003 and 0.015 $\mu g$ $m^{-3}$. EBC, $SO_4^2$, $NO_3^-$ and $NH_4^+$ are typical anthropogenic tracers during the Arctic haze season where the highest levels are observed. Their dominant sources in the mid-latitudes are fossil fuel combustion (EBC, $SO_4^{2-}$, $NO_3^-$), residential heating (EBC) and agriculture ($NH_4^+$) (AMAP, 2015). With regards to EBC, Tiksi (71°N) and Kevo (69.4°N) experience the highest concentrations and the largest variability, likely a result of their proximity to both anthropogenic source regions and potential local influences as well. The annual cycles are similar for all stations showing the typically elevated concentrations during the haze period. However, Zeppelin experiences relatively lower concentrations in January and February. The difference among these stations might be due to the comparison of different years and not necessarily

physical aerosol processes. At Summit, EBC peaks later, which has a different seasonal pattern, notably the absence of winter haze, due to its elevation.

Alert, Zeppelin, Barrow / Utqiagvik and Villum are comparable in their $SO_4^{2-}$ concentrations and variability. The four stations show almost identical annual cycles with the characteristic haze peak in late winter/early spring and the minimum in summer. Kevo's annual cycle is very similar to the aforementioned stations, however absolute concentrations are about a factor three higher. This difference is partly because of its relative vicinity to emission sources, and partly because the time series goes back to the 1960s when concentrations were higher in general. Thule and Gruvebadet show a more pronounced relative increase in April. $NH_4^+$ and $SO_4^{2-}$ are very similar in their annual cycles, which is consistent with the formation of ammonium sulfate or ammonium bisulfate. At Barrow / Utqiagvik $NH_4^+$ concentrations are roughly twice as high as at Alert, Zeppelin and Villum, indicating higher availability of $NH_4^+$ and less acidic particles since the $SO_4^{2-}$ concentrations are similar at all three stations. The box and whisker plots at Villum, Zeppelin and Alert are similar. $NO_3^-$ concentrations are higher than those of $NH_4^+$ at Villum and Zeppelin and variable. This result could be related to atmospheric chemical processes of particulate $NO_3^-$ formation, which depends on the aerosol acidity, water content and ambient temperature (see the discussion of the long-term trends for more details).

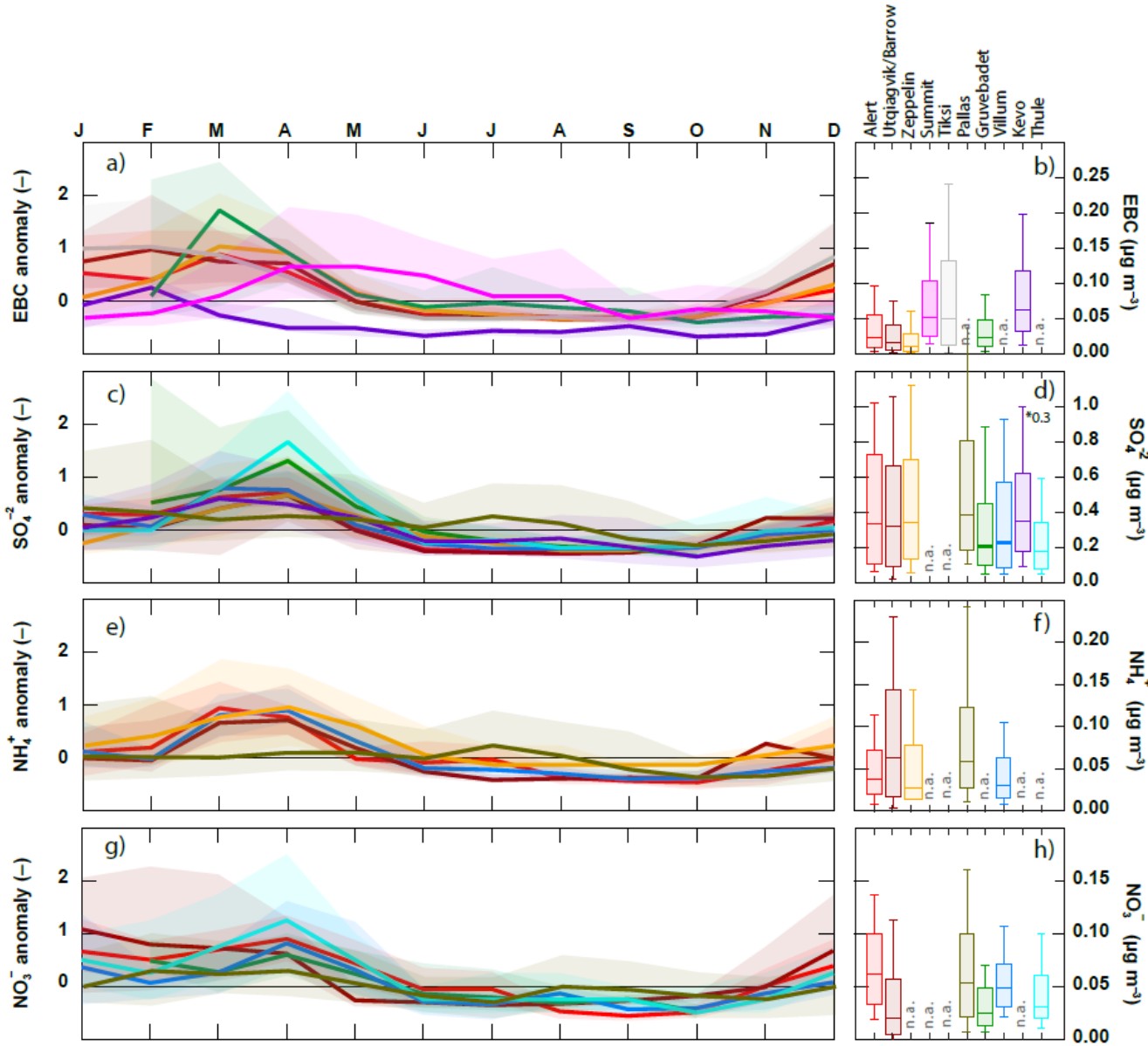

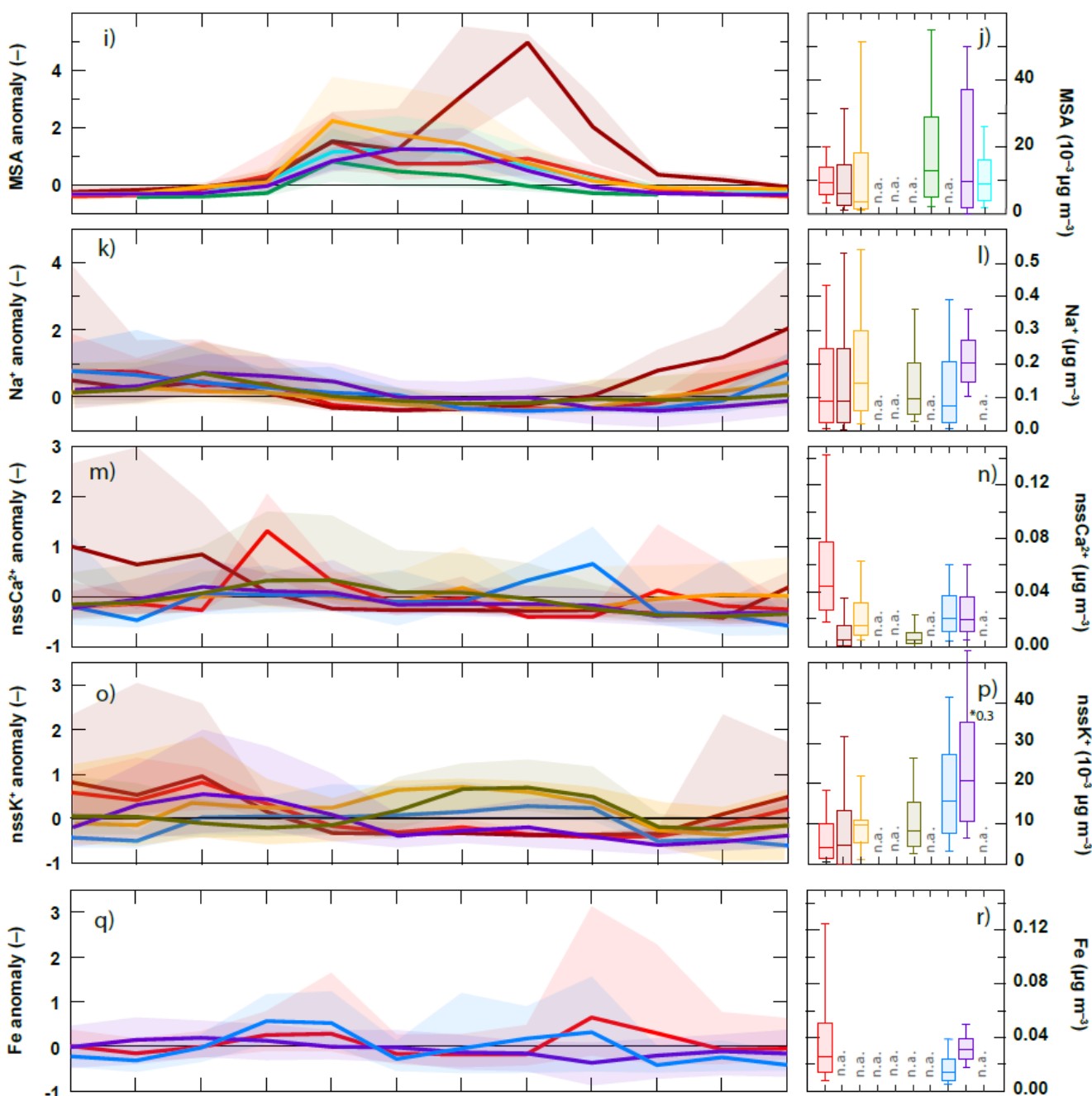

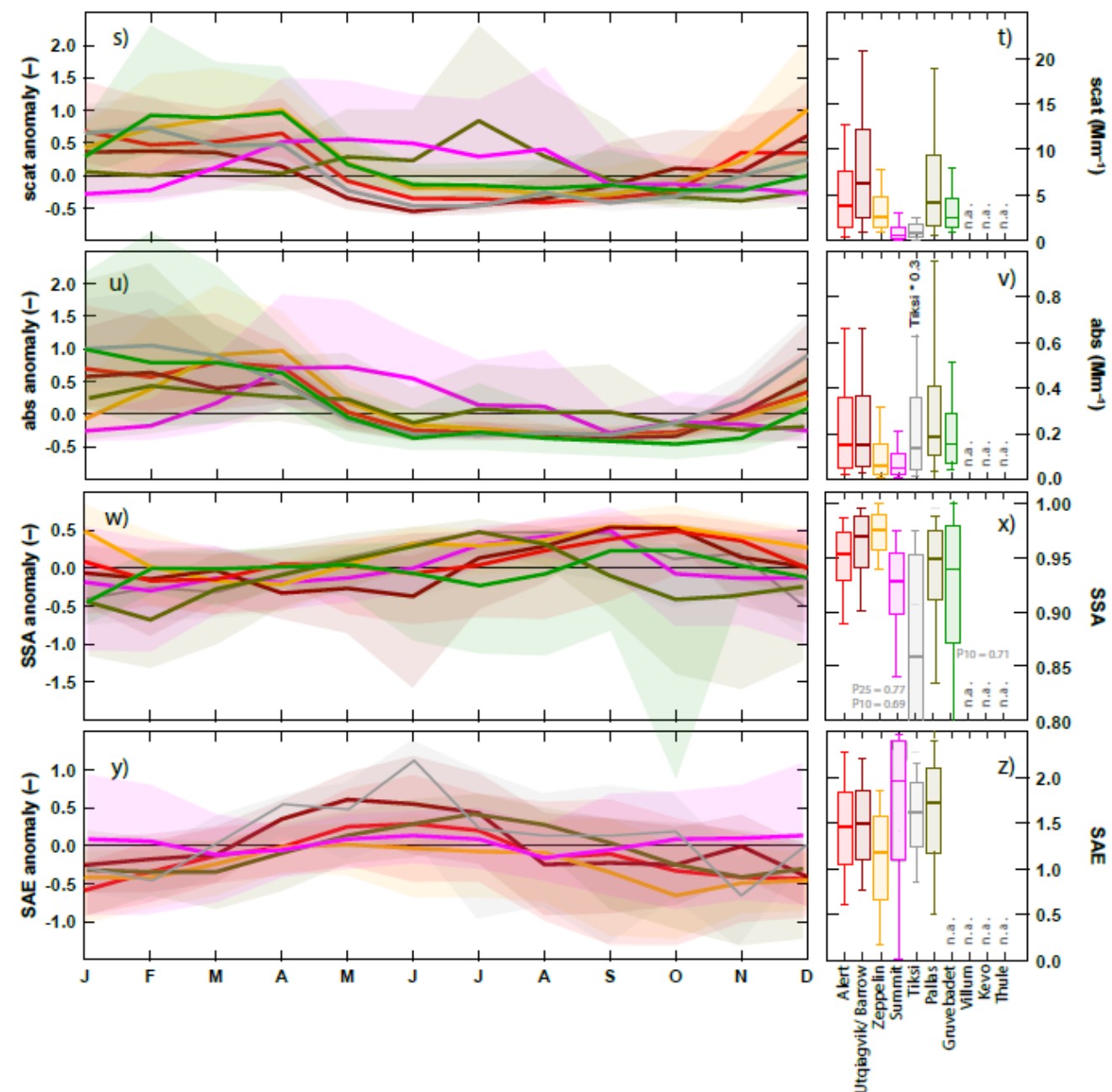

**Figure 2: The left column shows annual cycles of aerosol chemical compounds and physical properties for each available data set as anomaly. The data are normalized by the median and interquartile range (see Methods). Colors represent stations given in the right column of the figure (see panel b). The right column provides statistical values (median, interquartile range and 10th and 90th percentiles) per variables and station. Chemical compounds were derived from the following size**

**ranges (see also Table 1): TSP for Alert and Villum, $PM_{10}$ for Zeppelin, and $PM_1$ for Barrow / Utqiagvik.**

The other chemical tracers are less uniformly distributed throughout the year and there are significant differences among the stations. MSA is an oxidation product formed from DMS, which is derived from marine algae. It is hence associated with the presence of open water (or marginal sea ice zones), nutrients, sunlight and DMS producing algae as is reflected in the annual cycle. At all stations, a characteristic spring bloom peak in April-May is visible (Becagli et al., 2019; Moffett et al., 2020), except at Kevo where the peak occurs in June-July. In August, Barrow / Utqiagvik features a second and even higher peak, likely influenced by regional phytoplankton blooms (Moffett et al., 2020; Quinn et al., 2002). The other three stations do not show such strong signals and their highest MSA peaks are confined to spring, when southerly air masses transport MSA from productive waters in the mid-latitudes (Sharma et al., 2019).

Another natural contribution to aerosol is $Na^+$, which originates from sea spray and salt-laden blowing snow. The annual cycles at Alert and Barrow / Utqiagvik are similar with highest concentrations in winter, which is evidence of mixed contributions from blowing snow (Huang and Jaeglé, 2017) and southerly advection of sea spray (Quinn et al., 2002). Zeppelin shows lower enhancement during winter likely because of its elevation (476 m a.s.l). Villum features an annual cycle very similar to Alert, albeit with less enhancement in between November and December. Pallas shows a "flat" annual cycle, hence exhibiting less variability, potentially due to the geographic location, while Kevo exhibits a late spring peak.

$NssCa^{2+}$ and Fe have been used as mineral and soil dust tracer by Sharma et al. (2019) for Alert, where they show a spring and autumn peak. The spring peak has been interpreted as long-range transported dust from Asian deserts based on a Lagrangian model study (Groot Zwaaftink et al., 2016), whereas the autumn peak corresponds to minimum snow coverage and hence potential local soil dust sources. The latter could also be the case for Villum for $nssCa^{2+}$ and Fe where the main peak in $nssCa^{2+}$ occurs in autumn, but the actual source is difficult to pinpoint (Nguyen et al., 2013; Heidam et al., 2004). The peak in Fe at Villum during the haze season could stem from long-range transport of air masses from source regions in easterly

direction, where mineral dust emissions from Asian deserts may play a significant role. Barrow / Utqiagvik exhibits a strong $nssCa^{2+}$ enhancement between January and February, which has been interpreted as a signal of long-range transported dust from Eurasia (Quinn et al., 2002). At Zeppelin, Kevo and Pallas the annual cycles do not show any pronounced features. Overall, there is least similarity in $nssCa^{2+}$ between all stations compared to other tracers.

NssK$^+$ can be interpreted as biomass burning tracer originating from domestic wood burning, agricultural or forest fires contributing during different times of the year. At Alert and Barrow / Utqiagvik the annual cycle of nssK$^+$ is similar to that of typical anthropogenic tracers, e.g., EBC, and has been identified as a haze tracer at Barrow / Utqiagvik by Quinn et al. (2002). Based on carbon isotope measurements and Lagrangian dispersion modeling it has been shown that domestic wood burning is an important source of black carbon in winter (Stohl et al., 2013; Winiger et al., 2019). Hence, it is conceivable that nssK$^+$ could also be an indication of domestic wood burning. However, at Villum nssK$^+$ has been found to originate potentially also from fossil fuel burning (Nguyen et al., 2013; Heidam et al., 2004). At Kevo, the annual cycle of the tracer is most similar to $SO_4^{2-}$, and different from EC. However, previous research has shown that EC and nssK$^+$ are correlated and hence wood burning is a likely driver of the signal (Yli-Tuomi et al., 2003b). The seasonal cycle of nssK$^+$ at Villum shows a late summer enhancement from soil dust (Heidam et al., 2004). At Zeppelin nssK$^+$ remains enhanced throughout May to September. The presence of nssK$^+$ and its interpretation as biomass burning marker is in agreement with carbon isotope source apportionment by Winiger et al. (2019), who show an elevated level of 60 % biomass burning contribution to black carbon at the station. It has been shown for Zeppelin that long-range transport of agricultural fires can have a strong impact on the aerosol population in spring (Stohl et al., 2007). At Pallas, nssK$^+$ does not exhibit a typical Arctic haze pattern, but rather a peak in summer pointing towards wild fires (Mielonen et al., 2010).

### 3.1.2 Optical properties

Figure 2 also shows aerosol optical properties, which are tightly linked to the chemical composition and absolute aerosol mass for the scattering and absorption coefficients. Similar figures showing the absolute

values as annual cycles have been discussed in Schmeisser et al. (2018) for data between 2012 and 2014, and by Quinn et al. (2002) and Delene and Ogren (2002) for Barrow / Utqiagvik for 1997 to 2000. Here, we show all data available to this study (see Figure 3). The annual cycle of $\sigma_{sp}$ reflects the higher aerosol burden during the haze season and the low concentrations during summer for the observatories at Alert, Barrow / Utqiagvik, Zeppelin, Gruvebadet and Tiksi. The slight decrease of SSA at Gruvebadet in July and August is likely due to the impact of forest fire emissions, as suggested by higher aerosol concentrations of biomass burning tracers in this period of the year (Turetta et al., 2016; Feltracco et al., 2020). Pallas and Summit exhibit an opposite behavior with peak values in spring and summer, respectively. Pallas, being located relatively far south is much more influenced by Eurasian anthropogenic emissions being downwind (easterlies prevail) of significant emission sources (Aalto et al., 2002; Asmi et al., 2011) and also by biogenic emissions from the surrounding boreal forest (Hyvärinen et al., 2011; Tunved et al., 2006). Summit is located 3238 m a.s.l. and is therefore not affected by winter haze, which manifests itself in the lower layers of the atmosphere. It is however subject to long-range transport of biomass burning and anthropogenic emissions during spring and summer (Schmale et al., 2011; Thomas et al., 2017; Stohl et al., 2006). The $\sigma_{ap}$ is closely related to EBC (and correlated 1:1 for all stations where EBC was derived with a uniform mass absorption cross section, see Table 2, except at Kevo where a thermal-optical method was used) and both variables follow a similar annual cycle (Figure 2, panels a, u) with the maximum values during the haze period at Alert, Barrow / Utqiagvik, Tiksi, Gruvebadet and Zeppelin. Summit shows much lower $\sigma_{ap}$ values and follows a different annual cycle with a maximum in spring and summer, comparable to the observation of $\sigma_{sp}$. Here again, the influence of long-range transported biomass burning emissions as well as emissions from other anthropogenic sources that can reach Summit throughout spring and summer is evident as well as the absence of haze pollution in winter. SSA, an inherent aerosol property independent of the absolute abundance of aerosol derived for Alert, Barrow / Utqiagvik, Zeppelin, Gruvebadet, Tiksi, Pallas and Summit, peaks in autumn and reflects the minimum in the $\sigma_{ap}$ during that time of the year. At Pallas SSA peaks earlier in summer than at the other stations and also drops more strongly during winter, likely due to the anthropogenic influence in winter and the stronger boreal forest contribution in summer. The lower SSA at Summit in autumn compared to the other stations has been interpreted by Schmeisser et al. (2018) as a darker background aerosol due to

increased local air traffic from the station (despite a pollution sector quality-controlled data product). *SAE*, another intensive property, behaves similarly at all six stations (Alert, Pallas, Summit, Barrow / Utqiagvik, Tiksi, and Zeppelin) with a peak in spring lasting through early summer. Given that larger *SAE* values are indicative of smaller particle diameters, this behavior is in agreement with the annual cycle of the particle number size distributions as discussed e.g., in Croft et al. (2016) and Freud et al. (2017) due to the absence of larger submicron modes. During the *SAE* peak, more frequent new particle formation enhances the Aitken mode population, while regular wet deposition depletes the coarser particles acting as cloud condensation nuclei (Freud et al., 2017).

## 3.2 Trend calculations based on Mann-Kendall and Theil-Sen's Slope

Figure 3 and
Table 3 give an overview and summary of the calculated trends and their significance for all stations and parameters. The left part of the figure provides the significance of the overall trend as a numeric value and optically by a colored marker (green means significant at a p-value $< 0.05$, see Methods) split by haze (JFMA) and summer (JJAS) seasons. An upward or downward pointing arrow, only for significant trends, represents the direction of the slope and the value indicates the change per year in the unit of the variable as indicated in the leftmost column. To the right the duration of the measurements are shown as grey bars, and within each decade the marker and arrow indicate the decadal trends and their significance. Numerical values are given in SI Figure S1. Generally, the optical property records represent the last two decades and are shorter than chemical records at several stations, which date back about three decades. For MSA, the record across stations is fragmented, with only Alert, Kevo and Barrow / Utqiagvik providing decadal information.

Figure 3 — Data coverage and trends table.

| species | station | all years JFMA p-value | slope | all years JJAS p-value | slope |
|---|---|---|---|---|---|
| EBC µg m⁻³ | Alert | 0.00 ↓ | -2.61E-03 | 0.19 | |
| | Utqiagvik | 0.09 ↓ | -1.22E-03 | 0.17 | |
| | Summit | NA | | NA | |
| | Zeppelin | 0.03 ↓ | -1.05E-03 | 0.44 | |
| | Gruve | 0.79 | | 0.27 | |
| | Kevo | 0.00 ↓ | -5.71E-03 | 0.00 ↓ | -3.75E-03 |
| | Tiksi | 0.33 | | 0.85 | |
| SO₄²⁻ µg m⁻³ | Alert | 0.00 ↓ | -3.67E-02 | 0.09 ↓ | -7.50E-04 |
| | Utqiagvik | 0.57 | | 0.79 | |
| | Zeppelin | 0.00 ↓ | -1.50E-02 | 0.05 ↓ | -1.58E-03 |
| | Gruve | 0.83 | | 0.05 ↓ | 9.29E-03 |
| | Pallas | 0.00 ↓ | -3.12E-02 | 0.02 ↓ | -1.22E-02 |
| | Kevo | 0.00 ↓ | -3.46E-02 | 0.00 ↓ | -1.46E-02 |
| | Villum | 1.00 | | 0.45 | (1992 - 2002) |
| | Villum | 0.13 | | 0.02 ↓ | -6.86E-03 (2008 - 2017) |
| | Thule | 0.28 | | 0.65 | |
| NO₃⁻ µg m⁻³ | Alert | 0.00 ↑ | 1.38E-03 | 0.00 ↑ | 4.66E-04 |
| | Barrow | 0.59 | | 0.07 ↑ | 1.67E-04 |
| | Gruve | 1.00 | | 1.00 | |
| | Pallas | NA | | NA | |
| | Villum | 0.10 ↑ | 2.89E-03 | 0.14 | |
| | Thule | 0.30 | | 0.70 | |
| NH₄⁺ µg m⁻³ | Alert | 0.00 ↓ | -3.02E-03 | 0.70 | |
| | Utqiagvik | 0.97 | | 0.16 | |
| | Zeppelin | 0.20 | | NA | |
| | Pallas | 0.00 ↓ | -4.39E-03 | 0.02 ↓ | -2.36E-03 |
| | Villum | 0.09 ↓ | -1.42E-03 | 0.55 | |
| MSA µg m⁻³ | Alert | 0.02 ↓ | -3.29E-04 | 0.00 ↓ | -2.80E-04 |
| | Alert | 0.79 | | 0.14 | |
| | Utqiagvik | NA | | 0.79 | |
| | Zeppelin | 0.17 | | 0.55 | |
| | Gruve | 0.14 | | 0.14 | |
| | Kevo | 0.15 | | 0.05 ↑ | 3.26E-04 |
| | Thule | NA | NA | 0.60 | |
| Na⁺ µg m⁻³ | Alert | 0.36 | | 0.97 | |
| | Utqiagvik | 0.57 | | 0.97 | |
| | Zeppelin | 0.98 | | 0.33 | |
| | Pallas | 0.73 | | 0.03 ↓ | -1.37E-03 |
| | Kevo | 0.00 ↓ | -3.04E-03 | 0.00 ↓ | -1.70E-03 |
| | Villum | 0.37 | | 0.75 | |
| nssCa²⁺ µg m⁻³ | Alert | 0.42 | | 0.00 ↓ | -2.15E-03 |
| | Utqiagvik | 0.38 | | 0.38 | |
| | Zeppelin | 0.82 | | 0.58 | |
| | Pallas | 0.00 ↓ | -6.21E-04 | 0.03 ↓ | -2.70E-04 |
| | Kevo | 0.00 ↓ | -1.10E-03 | 0.00 ↓ | -6.72E-04 |
| | Zeppelin | 0.94 | | 0.52 | |
| | Villum | 0.76 | | 0.81 | |
| nssK⁺ µg m⁻³ | Alert | 0.10 | | 0.97 | |
| | Utqiagvik | 0.82 | | 0.15 | |
| | Pallas | 0.07 ↓ | -3.72E-04 | 0.63 | |
| | Kevo | 0.00 ↓ | -1.61E-03 | 0.00 ↓ | -1.36E-03 |
| | Zeppelin | 0.87 | | 0.57 | |
| | Villum | 0.33 | | 0.22 | |
| Fe µg m⁻³ | Alert | 0.00 ↓ | -4.73E-04 | 0.02 ↓ | -5.54E-04 |
| | Villum | 0.27 | | 0.11 | |
| | Kevo | 0.02 ↓ | -2.65E-04 | 0.53 | |
| Scat Mm⁻¹ | Alert | 0.12 | | 0.11 | |
| | Utqiagvik | 0.00 ↓ | -2.41E-01 | 0.63 | |
| | Zeppelin | 0.01 ↑ | 1.79E-01 | 0.00 ↑ | 5.48E-02 |
| | Gruve | 0.13 | | 0.94 | |
| | Summit | 0.46 | | 0.62 | |
| | Tiksi | NA | NA | NA | NA |
| | Pallas | 0.14 | | 0.24 | |
| Abs Mm⁻¹ | Alert | 0.09 ↓ | -1.50E-02 | 0.38 | |
| | Utqiagvik | 0.00 ↓ | -1.37E-02 | 0.56 | |
| | Zeppelin | 0.02 ↓ | -7.84E-03 | 0.80 | |
| | Gruve | 0.79 | | 0.27 | |
| | Summit | 0.14 | | 0.32 | |
| | Pallas | 0.39 | | 0.71 | |
| | Tiksi | 0.28 | | 0.77 | |
| SSA | Alert | 0.81 | | 0.81 | |
| | Utqiagvik | 0.05 ↑ | 9.18E-04 | 0.24 | |
| | Zeppelin | 0.11 | | 0.01 ↑ | 2.82E-03 |
| | Gruve | | | | |
| | Pallas | 0.39 | | 0.39 | |
| | Tiksi | NA | | NA | |
| | Summit | 0.88 | | 0.14 | |
| SAE | Alert | 0.10 ↓ | -3.65E-02 | 0.11 | |
| | Utqiagvik | 0.90 | | 0.07 ↓ | -1.32E-02 |
| | Pallas | 0.75 | | 0.26 | |
| | Tiksi | NA | | NA | |
| | Zeppelin | 0.00 ↓ | -6.63E-02 | 0.00 ↓ | -8.43E-02 |

**Figure 3: Data coverage and trends.** The grey bars on the right side indicate data availability per station and variable. The p-values and slopes on the left side indicate the overall trends

characteristics per station and variables separated by season (JFMA and JJAS). Round markers indicate the significance of the trend, where green corresponds to a p-value < 0.05, orange to 0.05 < p-value < 0.10, and red p-value > 0.10. The arrows indicate whether the trend is positive or negative. Decadal trend significance and direction are given by markers only in the respective time period. The two left (right) symbols correspond to JFMA (JJAS). Chemical compounds were derived from the following size ranges (see also Table 1): TSP for Alert and Villum, $PM_{10}$ for Zeppelin, and $PM_1$ for Barrow / Utqiagvik. NA means there are not enough values above detection limit to calculate a p-value.

**Table 3: Summary of derived significant trends per season (JFMA and JJAS) for the full period of each data set and per decade. The percentage of significant (sign.) trends is calculated from the total number of trends (column 2) evaluated for the given time period. There are 26 significant decadal trends.**

| period | no. of trends | total sign. | JFMA | | | JJAS | | |
|---|---|---|---|---|---|---|---|---|
| | | | sign. pos. | sign. neg. | percent sign. | sign. pos. | sign. neg. | percent sign. |
| full period | 73 | 41 | 3 | 20 | 30.1% | 5 | 14 | 26.0% |
| 1970 | 6 | 1 | 1 | 0 | 16.7% | 0 | 0 | 0.0% |
| 1980 | 16 | 1 | 0 | 0 | 0.0% | 0 | 1 | 6.3% |
| 1990 | 26 | 5 | 0 | 4 | 15.4% | 0 | 1 | 3.8% |
| 2000 | 41 | 10 | 2 | 2 | 9.8% | 3 | 3 | 14.6% |
| 2010 | 51 | 9 | 1 | 2 | 5.9% | 3 | 3 | 11.8% |

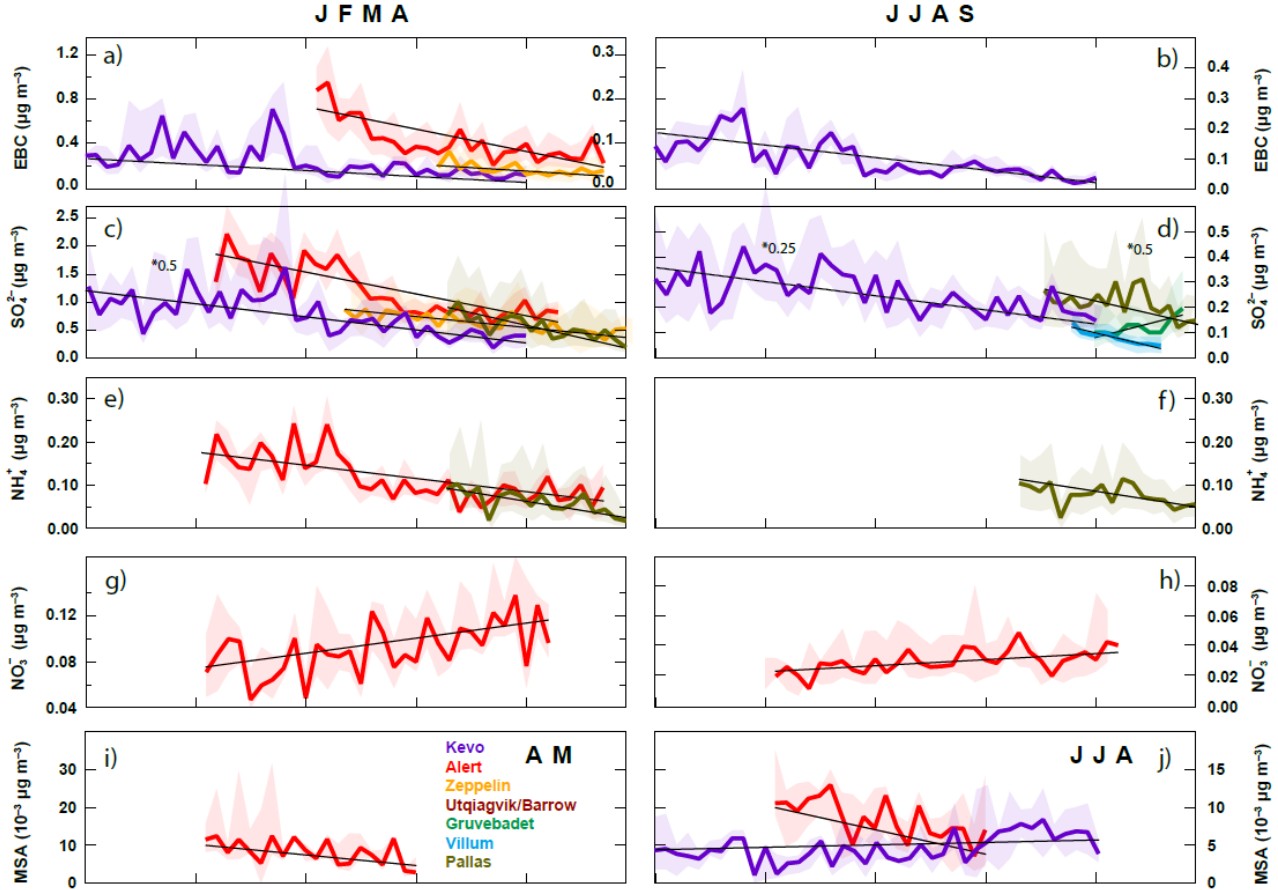

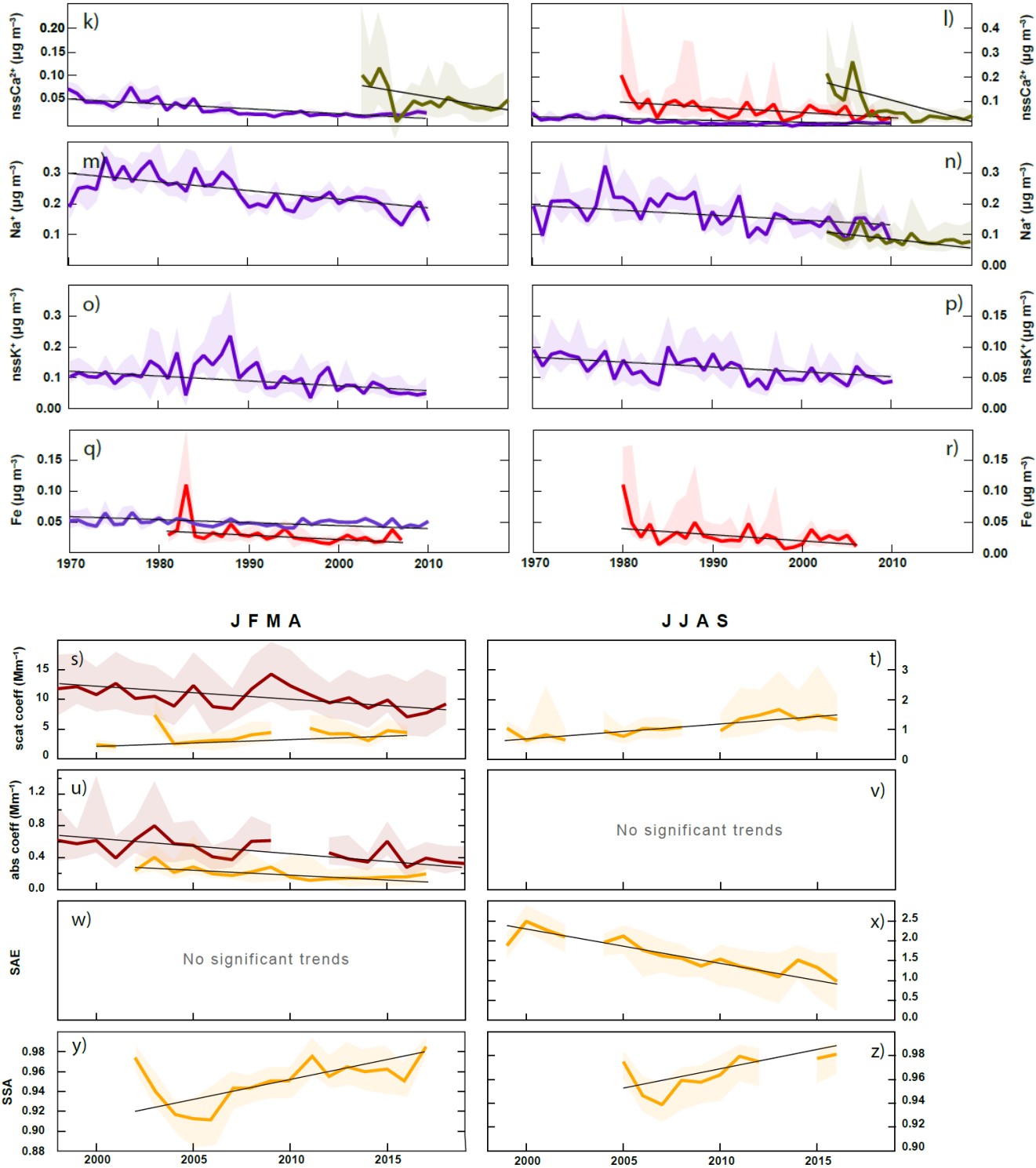

**Figure 4: Significant trends (p < 0.05) per variable and station for the entire time period over which data are available. For Villum we considered the time period 2008-2016 here. MSA is given for the months of April-May and June-July-August.**


### 3.2.1 Long-term trends of $SO_4^{2+}$, $NH_4^+$, $NO_3^-$ and EBC

For the full period of each measurement, during the haze season 30 % of the trends are significant ($p <$ 0.05) with 20 negative, and three positive trends. For the summer, 26 % of overall trends are significant

with 14 negative and five positive trends. We show all 41 significant trends in Figure 4. In winter, typical haze tracers such as EBC, $SO_4^{2-}$ and $NH_4^+$ make up nine of the 20 significant and full period negative trends at Alert, Zeppelin, Pallas and Kevo. This is consistent with the reduction of air pollution emissions in North America, Europe and Russia, which we show in Figure 5 based on the ECLIPSEv6b (Höglund-Isaksson et al., 2020) and CMIP6 emission data between 1970 and 2014 (Hoesly et al., 2018). For BC,

the emissions data show a gradual decrease from 1970 to 2000 for the Western Arctic (Western Arctic refers to the USA and Canada), while emissions in the Eastern Arctic (Eastern Arctic refers to the Kingdom of Denmark, Finland, Iceland, Norway, the Russian Federation, and Sweden) and the rest of Europe show a strong decrease between 1990 and 2000. Asia (Japan, People's Republic of China, Republic of India, Republic of Korea, Republic of Singapore) shows a decrease in the 1990s and a

subsequent increase since the early 2000s. The EBC trend at Kevo reflects the sharp decline in 1990, and also at Alert most of the EBC concentration decline happened before 2000, reflecting primarily the economic change in the former Soviet Union (Dutkiewicz et al., 2014; Sharma et al., 2019). However, a significant decadal winter trend is only visible for Alert and Barrow / Utqiagvik in the 1990s. Otherwise, decadal trends are not significant. After the year 2000 emissions slightly declined in the Eastern Arctic

and rest of Europe. In the Western Arctic they slightly increased towards 2005 and then decreased towards 2008 while then staying at the same level. In Asia emissions were on the rise until 2008 and then declined, while in the rest of the world emissions increased. Arctic EBC concentrations, which are mostly impacted by the Eastern and Western Arctic as well as Europe (Backman et al., 2021), neither show a decline nor incline from 2000 onwards during winter.

To demonstrate the influence of short-term variability in the last decade on the significance of trends, we tested the required yearly rate of increase or decrease in EBC concentrations to detect a significant trend (p-value < 0.05) within the last ten or close to ten years of measurements at the following stations: Alert (2007 - 2017), Zeppelin (2007 - 2017), Barrow / Utqiagvik (2009 – 2019), Gruvebadet (2011 – 2018), and Tiksi (2010 – 2018). We calculated the trend line for the years given in parenthesis, de-trended the

EBC concentration data with it and then added $x_i$ % of EBC increase per year in steps of 1 (i.e. $x_1 = 1$ %, $x_2 = 2$ % and so on) until the trend emerged as significant. The results for Alert and Zeppelin are 5 % yr$^{-1}$, for Barrow / Utqiagvik 6 % yr$^{-1}$, for Tiksi 11 % yr$^{-1}$, and for Gruvebadet 16 % yr$^{-1}$. Hence, the present-day short-term variability requires relatively large changes to occur to derive significant trends with the Mann-Kendall Theil-Sen method for time periods less than 10 years. This explains why trends on the

order of 1% or less per year cannot be significant over single decades but are significant over a record of 20 or more years (as for the case of EBC at Zeppelin in winter, see Figure3). Therefore, even if none of the EBC trends in the haze season turned out to be significant for the 2010-2019 period (in contrast with previous decades), such "stagnation" must be considered carefully. In addition, detection of a 1 % change would not reflect large changes in air quality. The absence of trend detection over shorter periods also

highlights the importance of maintaining monitoring activities at the Arctic observatories in the long run, as decadal changes (occurring over periods of > 10 years) in atmospheric composition can be detected with more statistical certainty.

     $SO_2$ emissions, a precursor to particulate $SO_4^{2-}$, decreased strongly in Europe and the Western Arctic countries since 1980 due to clean air policies such as UNECE LRTAP and the US Clean Air Act. $SO_2$

emissions showed a clear dip between 1990 and 2000 in the Eastern Arctic, probably due to the economic change in the former Soviet Union. Elsewhere emissions increased strongly, except for the late 1990s in Asia. Winter concentration trends at Alert, Kevo and Zeppelin follow the pattern of the strong emission decline in the 1990, showing the strongest decline in this decade and then leveling off. There is only one significant decadal wintertime trend for $SO_4^{2-}$, which occurred at Alert in the decade of the strongest $SO_2$

emission reductions, 1990-2000. There are only few observations of $NH_4^+$, and only Alert and Pallas show a negative overall trend during the haze season. $NH_4^+$ forms ammonium bisulfate and, to a lesser extent, ammonium sulfate in the particle phase and hence follows the trend of $SO_4^{2-}$. This is also observed

at the other two stations that measured $NH_4^+$, Barrow / Utqiagvik and Villum, but the p-values do not indicate significance for the overall period ($0.05 <$ p-value $<0.1$).

$NO_3^-$ shows an opposing significant trend, i.e. positive, at Alert compared to $SO_4^{2-}$.. The favored partitioning into the particle phase is likely the main reason, also for the similar trend observed during summer at Alert. It is important to keep in mind that Alert is the only site where a measurement record longer than 30 years has been acquired. The positive trend at Alert is opposite to the emission reduction efforts of $NO_x$ (see Figure 5), the precursor to $NO_3^-$. However, with the reduction in particle acidity, mainly

caused by lower emissions of $SO_2$ and subsequent formation of particulate sulfate, more particulate nitrate can be formed (Sharma et al., 2019). In addition, cold temperatures and high particle water content will favor the partitioning of nitrate into the aerosol phase (Guo et al., 2016; Shah et al., 2018). The analysis of $NO_3^-$ clearly shows that not only emission changes are relevant for chemical trends of Arctic aerosols, but also the type of dominating chemical reactions that change due to an altered chemical composition of

the atmosphere.

Some of the above discussed tracers also show significant trends during summer for the overall period: five negative and two positive. EBC decreased at Kevo, here again with almost a step change in the 1990s, due to the site's proximity to the former Soviet Union which collapsed in the 1990s. Sulfate decreased significantly at Kevo, Pallas between 2005 and 2020, and Villum over the period from 2008 to 2017.

Enhanced microbial activity during the summer has been hypothesized to be a responsible factor for particle mass formation at Villum (Dall´Osto et al., 2018), even though the summer trend is negative. However, the aforementioned study focused on the smallest particles, while the filter based determination of $SO_4^{2-}$ targets the accumulation mode and even larger particles. Similar new particle formation studies have been conducted for Zeppelin and Gruvebadet (Becagli et al., 2019; Dall´Osto et al., 2017; Beck et al.,

2020; Dall'Osto et al., 2019), where only at Gruvebadet $SO_4^{2-}$ trends are positive and significant during summer. A plausibility check, on whether marine microbial activity is the driving factor, could be done using MSA concentrations. While those are not available at Villum or Gruvebadet, no significant trend in MSA has been found at Zeppelin station, close to Gruvebadet. The absence of an MSA trend might be an indication that reaction pathways are too complex to expect a directly measurable concentration effect

in both particulate $SO_4^{2-}$ and MSA based on microbial activity. Also note that Villum is sea ice dominated

for ten months of the year. Mahmood et al. (2019) found that enhanced microbial emissions can lead to higher CCN number concentrations which implies a subsequent removal of the corresponding emission products by wet deposition and hence might not be observed at permanent stations. The increase in $SO_4^{2-}$ around Svalbard is likely not due to increased shipping activities, e.g. from tourism. Eckhardt et al. (2013) show a strong influence in summer on sulfur from ships directly in Kongsfjorden, but on an annual basis this is a small effect. Moreover, recently new emission regulations require low sulfur-content fuel. Generally, observatories that started measurements only after the year 2000, detect neither significant haze nor summer trends for anthropogenic tracers, because most of the air pollution reduction happened prior to that.

### 3.2.2 Long-term trends of MSA, Na$^+$, nssK$^+$, nssCa$^{2+}$, and Fe

The other chemical constituents, are responsible for 16 significant trends, seven in winter and nine in summer. All are negative, except for MSA at Kevo. Focusing on MSA first, at Kevo concentrations started increasing since the 1980s. The significant positive trend has been identified before by Laing et al. (2013), who found significant correlations with both increasing sea surface temperature in the source regions (Barents, Norwegian and Greenland Seas) and declining sea ice coverage. Warmer sea water and less sea ice are generally interpreted as leading towards higher marine microbial productivity. Interestingly, at Alert, the concentrations of this ion show negative trends for the haze and summer seasons between 1980 and 1999. Here, the haze season concentrations were related to long-range transport from more southerly latitudes, while the summer season concentrations were related to more regional Arctic ocean sources (Sharma et al., 2019). Hence reasons for the two seasonal trends might be different and could reflect both variability in marine biological activity as well as long-range transport patterns and removal processes on route. Within the decade 2000 to 2010, the summer trend turns significant and positive as discussed in Sharma et al. (2012).The overall trend of MSA at Barrow / Utqiagvik in the haze season is negative for the period between 1998 and 2012 (not shown for reasons discussed below). The summer trend is not significant over this period at this station. Conversely, concentrations increased significantly at Barrow / Utqiagvik during the summer season between 1999 and 2007 as discussed in Quinn et al. (2009). Note that they used July through September for the summer months, while here we use June through August.

Calculating trends over the decade 2000 to 2010 hence results in no significant trend at Barrow / Utqiagvik (p-value of 0.09 and negative slope). This seeming contradiction is an important point, because the newly available longer time series shows that Quinn et al. (2009) described a development over a period which happened to show an increase, but which is not necessarily indicative of the longer-term trend. Recently the Barrow / Utqiagvik MSA time series was extended by Moffett et al. (2020) who found an overall positive trend for July through September between 1998 and 2017, further underlining the argument that trend calculations over different periods can lead to different results. This is particularly true for MSA concentrations which do not seem to follow a monotonic long-term trend, but instead show substantial variability over time scales of five to 15 years, which renders the Mann-Kendall approach inappropriate for further investigation.

$Na^+$ can also be regarded as a natural tracer, i.e. for sea spray its dominant source. No significant positive trend is observed, despite the fact that the potential for more sea spray aerosol formation due to retreating sea ice is often referred to in the literature (Browse et al., 2014; Struthers et al., 2011). However, not observing increased $Na^+$ concentrations does not mean that there is no enhanced sea spray production. Sea salt particles are efficient CCN and hence may preferentially be removed from the atmosphere by wet deposition as has been shown for other hygroscopic substances (Mahmood et al., 2019), or air mass transport patterns can change (Heslin-Rees et al., 2020).

$NssCa^{2+}$ shows negative trends at Kevo and Pallas during winter and summer. Given the stations' vicinity to industrial emissions in the east and south-east, which have been identified as source regions previously (Yli-Tuomi et al., 2003a; Laing et al., 2014b) the decline could be related to changing industrial activities, but warrants more detailed investigation. $NssCa^{2+}$ has also been interpreted as a soluble tracer for mineral and soil dust, and shows a negative trend at Alert in summer, which is consistent with the results by Sharma et al. (2019). There is no evident explanation for the decrease, particularly given the decreasing snow cover and potential local sources of dust in late summer (see discussion of annual cycles). Sharma et al. (2019) hypothesize that long-range dust transport could play a role in this context.

Iron is another dust tracer and exhibits a similar annual cycle compared to $nssCa^{2+}$ at Alert, and the summer trend is also negative supporting the hypothesis of similar sources and or processes. However, the haze trend of Fe is also negative, while the $nssCa^{2+}$ trend is not significant, indicating that, at least

during winter, sources might be different for the two constituents. The shape of the long-term concentration curve of Fe is similar to that of EBC, $SO_4^{2+}$ and $NH_4^+$, with the strongest decline in the mid-1990s. Given that Fe is emitted in fossil fuel combustion in addition to originating from soil dust (Matsui et al., 2018), this observation points towards an anthropogenic contribution to Fe observed at Alert. At Kevo, the Fe long-term trend is also negative. Even though Fe trends have not specifically been investigated, Laing et al. (2014a, b) looked at source regions and long-term trends of trace metals at Kevo and found a general significant decline due to changed industrial activities. It is likely that Fe stems from the same or related sources, because the seasonal cycle is similar to the investigated trace metals.

Only at Kevo, significant and negative trends for $nssK^+$ were found for both the haze and summer seasons. As discussed for the seasonal cycles, this tracer is likely related to decreased domestic wood burning. Given the peak during the haze season, typical for anthropogenic emissions, and the similarity to the EC trend, the source could additionally be related to industrial activity and its decline in the source regions (Dutkiewicz et al., 2014). However, this has not explicitly been investigated.

Given the recent increase in fire activity in the Arctic, one might have expected an emerging trend in $nssK^+$ as biomass burning tracer. Reasons for the absence of a general upward trend may be that biomass burning emissions are carried further aloft in the troposphere and are not well captured at the surface, and that station records do not cover the last years, where the strongest emissions in consecutive years occurred (McCarty et al., 2021).

### 3.2.3 Long-term trends of optical properties

Aerosol optical properties, measured in situ at the surface, have also changed significantly at several stations during the last decades. Significant trends are observed in the scattering coefficient at Barrow / Utqiagvik and Zeppelin during winter, and at Zeppelin during summer with regard to the overall time period. At Barrow / Utqiagvik, the trend is negative during the haze period for all observations (1998-2019), and for both haze and summer seasons in the recent decade (2010-2019). Note that these overall and recent decreasing trends in scattering coefficient during winter are consistent with Collaud Coen et al. (2020b). However, they did not find a significant trend in scattering coefficient during the summer months at Barrow / Utqiagvik for the most recent decade, though they used a different seasonal definitions

as well as different time aggregations. The scattering coefficient can be interpreted as a measure of the total measured aerosol concentrations and thus provides evidence of a decline in overall particulate matter, if hygroscopic growth or swelling is prevented through drying particles before measurements, which is the case at the discussed sites. At most stations, $Na^+$, $SO_4^{2-}$ and organics make up most of the mass (Petäjä et al., 2020) and are also known to scatter light rather than absorb it. A direct comparison with the aerosol mass is difficult for Barrow / Utqiagvik, because most of the chemical and optical data only overlap between 1997-2013. However, using linear regression Quinn et al. (2002) have shown for data between 1997 and 2000 that $Na^+$ exerts a dominant role on the winter aerosol scattering coefficient, while $SO_4^{2-}$ dominates the spring scattering. Aerosol filter samples have been taken throughout the full period (1997-present), but data were not available beyond 2013 for this work. Future analysis will help answer the question whether an anthropogenic or natural source, or both, contribute to the observed decline. Conversely, at Zeppelin the overall summer and winter trends are positive for $\sigma_{sp}$, and strongly driven by changes in the decade between 2000 and 2010. The *SAE* and $\sigma_{sp}$ trends have been discussed in detail by Heslin-Rees et al. (2020) who found that the increased scattering signal is likely due to an increased presence of coarse mode particles. This is reflected by the significant negative trend in the *SAE* (also for both seasons in their publication). While it could be hypothesized that the retreat of sea ice is responsible, Heslin-Rees et al. (2020) found that a change in the atmospheric circulation, in which increased advection of air masses from the southwest, i.e. the Atlantic, can explain the observations. The conclusion is that more sea spray aerosol is transported towards Zeppelin contributing to the increased coarse mode particles. This is not confirmed by the derived trends of sea-salt particles with the same method ($Na^+$, see previous section). Here trends in $Na^+$ were not found to be significant for the Zeppelin station. However, the physico-chemical characterization of coarse-mode aerosols at Gruvebadet (Song et al., 2021; Udisti et al., 2016) showed the occurrence of several populations of anthropogenic, sea-salt and crustal particles exhibiting distinct coarse mode size-distributions, which complicates the attribution of trends in *SAE* to individual changes in emissions.

The absorption coefficient, which is mostly related to the presence of combustion-related aerosol and, to a much lesser extent, to mineral dust, exhibits negative trends for the overall time periods of observations at Barrow / Utqiagvik and Zeppelin in winter, consistent with the wintertime trends for absorption

coefficient reported in Collaud Coen et al. (2020b). These trends are in line with the decreasing EBC concentrations during the haze season, where the EBC trend is significant at Zeppelin, while at Barrow / Utqiagvik the p-value is 0.09.

Pallas shows a negative decadal trend for *SAE* between 2000 and 2009 in summer. Collaud Coen et al. (2013) and Lihavainen et al. (2015) showed a significant negative trend for that period in winter, winter although no significant trend is reported in Collaud Coen et al. (2020b). Thereafter variability has become larger between the years and the amplitude of the signal also increased, for which no particular reason has been found. In contrast, Collaud Coen et al. (2020b) observed both increasing and decreasing statistically significant *SAE* trends at various Arctic sites depending on the period and season considered.

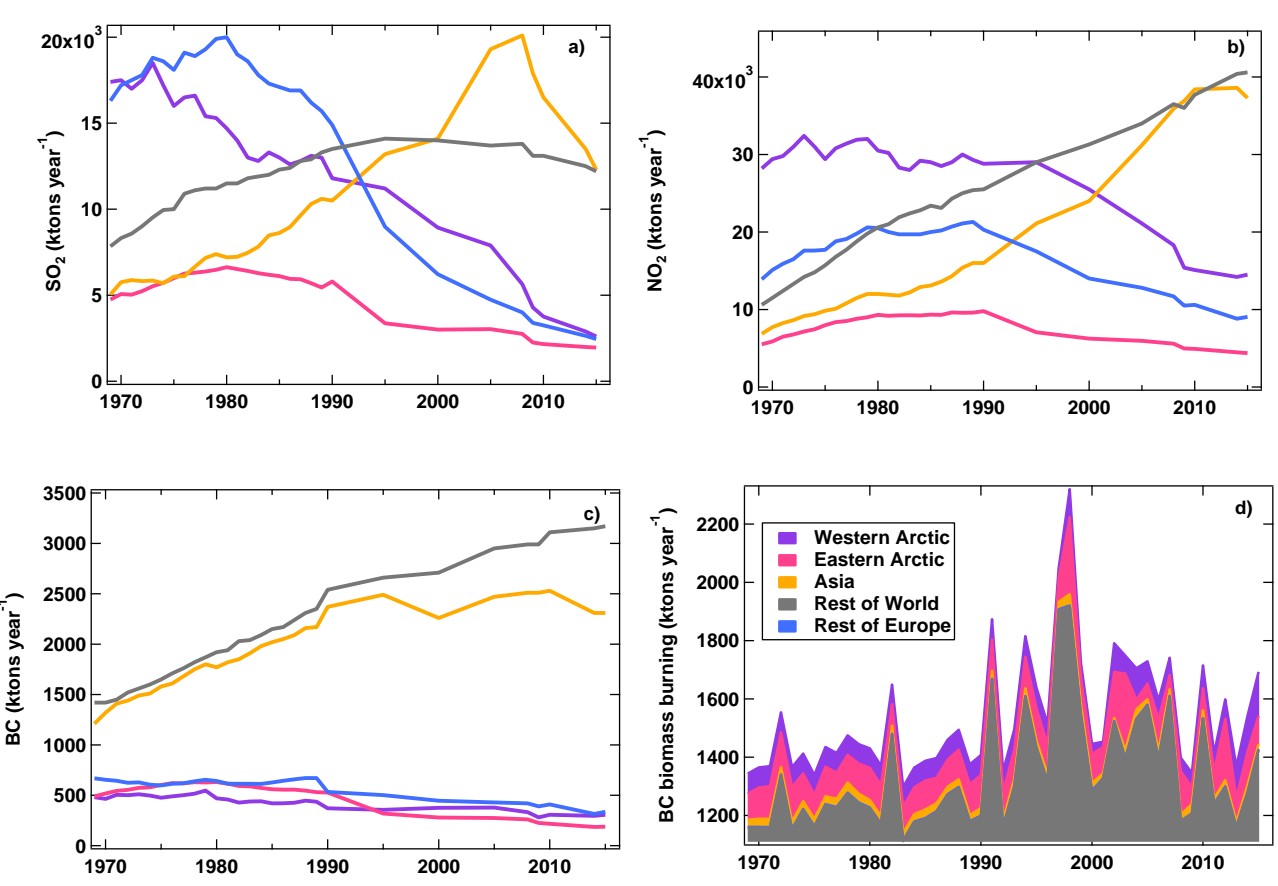

**Figure 5: Anthropogenic emissions of SO$_2$ (a), NO$_2$ (b), BC (c), and BC from biomass burning (d). Anthropogenic emissions are based on ECLIPSE v6B (Höglund-Isaksson et al., 2020) . Data from**

before 1990 are based on CMIP6 (Hoesly et al., 2018; van Marle et al., 2017) scaled linearly to match ECLIPSE v6B. Biomass burning data are from CMIP6. The Western Arctic region refers to the following countries: USA and Canada; Eastern Arctic region: Kingdom of Denmark, Finland, Iceland, Norway, the Russian Federation, and Sweden. The grouping follows the report of the Arctic Monitoring and Assessment Programme on short-lived climate forcers and refers to entire countries (AMAP, 2021).

## 4 Discussion

The results presented above reflect the effort, which has been made to characterize the aerosol chemical and optical properties across the Arctic over several decades, and they highlight the importance of this long-term effort. Importantly, to obtain a pan-Arctic view, records must be taken over the same period of time. Based on the data set behind this study, in the 1980s, on average, measurements of 16 variables (chemical and optical) were carried out each year at any station considered here. In the 1990s the number increased to 31, and further to 58 variables in the 2000s. In the latest decade, annual measurements of 66 variables were available on average. Note that not all data might have been released for the most recent years and hence the number is a lower estimate. We also limited the selection of aerosol variables measured to meet the objectives of the presented study (long-term trends).

The data allow us to draw conclusions on how changes in emissions, processes and transport in the northern hemisphere have changed the aerosol properties across the Arctic. Note that the applied methodology here, i.e. the Mann-Kendall significance test and Theil-Sen slope estimator, relies on monotonic change. Hence, time series which do not change monotonically are less likely to exhibit a significant trend (see e.g. the MSA concentration at Alert). We find in total 41 significant trends over full records, i.e. spanning more than a decade, compared to 26 significant decadal trends (see Table 3). The majority of significantly declining trends in anthropogenic tracers occurred during the haze period, and was driven by changes in the 1990. Even though the decade from 1990 to 1999 stands out, only five of the 26 trends calculated for this decade are significant. This result means that long-term observations that cover multiple decades are extremely valuable and necessary to document the anthropogenic influence on the Arctic aerosol concentration. In addition, it can be concluded that observations which started after the 1990s cannot reflect the most important decline of the anthropogenic aerosol concentration in the Arctic in the recent past. Nevertheless, the more recently established

monitoring activities will provide valuable insight into recent trends in the years to come with respect to anthropogenic emission changes within (Kolesar et al., 2017; Aliabadi et al., 2015; Schmale et al., 2018) and outside of the Arctic, as well as to potential changes in natural emissions based on climate feedback mechanisms. For example, we find from the recent records that EBC concentrations appear to be stagnating, i.e., no significant change is observed. Because trends in the latest decade are not significant, continuing observations in the future is critical to understand the present-day development. The newly established time series at additional geographical locations will help evaluate this trend in the future; particularly in light of the current black carbon reduction policies, such as those initiated by the Arctic Council countries through the Fairbanks Declaration 2017 (Arctic Council, 2017), and whether those policies will have an effect across the Arctic.

Anthropogenic influence has not been restricted to the haze season. Several significant negative trends in summer for EBC, $SO_4^{2-}$ and Fe seem to demonstrate this, because there are corresponding negative trends during the haze period as well. With the decline of anthropogenic air pollution, natural aerosol components emerge more strongly now in summer than previously and are hence responsible for the larger part of the summertime aerosol climate effects, including both the direct and indirect effects. The vanishing anthropogenic summer influence coincides with rapid and unprecedented environmental change in the high northern latitudes. September Arctic sea ice is retreating by 13.1 % per decade since 1979 compared to the 1981-2010 average (Arctic Sea Ice Minimum, 2020). Sea ice retreat creates more open ocean, which creates larger fetch for wave development (Casas-Prat and Wang, 2020) and hence more sea spray emission. DMS emissions have increased by 33 % per decade in ice-free water since 1998 (Galí et al., 2019). Boreal forest fires are also becoming more frequent and severe (Rogers et al., 2020). For example, BC emissions have roughly doubled from 2010 to 2020 from wildfires north of 60°N (McCarty et al., 2021). Warmer temperatures, which are the main cause of enhanced fire activity, also cause enhanced biogenic emissions, such as isoprene (a secondary organic aerosol precursor originating from vegetation (Arneth et al., 2016)). These and other ongoing and anticipated environmental changes (e.g., changed wind and air mass transport patterns) will have an effect on the natural contribution to aerosol properties and concentrations (Schmale et al., 2021). However, the trend analyses of natural aerosol tracers used in this study do not show strong or unambiguous evidence for such changes yet.

There can be several reasons. With respect to $SO_4^{2-}$, there could be compensating effects of reduced $SO_2$ emissions and increased DMS emissions. For example, a high interannual variability in natural sources hinders the detection of long-term trends. Additionally the complex relationships between natural emissions, atmospheric transport, chemical reactions, particle removal processes and observed airborne aerosol concentrations, means that an increase in emissions does not automatically translate to enhanced concentrations of corresponding end products or the species itself. Furthermore, our analyses are conducted with the Mann-Kendall Theil-Sen Slope test only. Other methods might yield different results. For example, increased wave breaking on a more ice-free ocean should lead to higher $Na^+$ concentrations. However, only two of the six stations that report $Na^+$ show a significant and negative trend in summer at Kevo and Pallas, and winter at Kevo. This could be due to the inherently large variability in the $Na^+$ signal, which requires hence longer data sets to extract a monotonic signal with the Mann-Kendall Theil-Sen methodology. For Alert, if the $Na^+$ concentration for the first five years of the record and the last five years are averaged, there is a 20 % increase in $Na^+$. It could also be due to the preferential wet scavenging of $Na^+$ due to its high hygroscopicity. In addition to the chemical signature, which can document changes in sea spray aerosol, size distribution data can provide indications and improve such analysis. Using the *SAE*, Heslin-Rees et al. (2020) showed an increase in coarse mode particles at Zeppelin, which is likely due to an increase of observations in sea spray emissions and atmospheric transport changes. At Zeppelin, the source is the mid-latitude Atlantic rather than the increasingly ice-free Arctic Ocean. While at Alert, the *SAE* trend was not found to be significant, it nevertheless has a low p-value of 0.09 and a negative slope, which is consistent with the decrease in $Na^+$.

Concerning marine biogenic emissions, observations of MSA concentrations provide mixed results. At Kevo the summer concentrations clearly increased since the 1980s. While at Barrow / Utqiagvik the MSA time series from 1997 to 2017 shows an increase (Moffett et al., 2020). Measurements at Alert do not indicate a significant upward trend between 1981 and 2018, but rather a negative trend between 1981 and 1999, and a positive trend in the decade 2000 to 2009 in MSA. Becagli et al. (2019) find a positive trend in the most recent decade at Gruvebadet on Svalbard and a negative one at Thule, Greenland. Our analyses in this study show similar trends for the two latter stations, but no significance. In a modeling study (Browse et al., 2014) found that enhanced marine biogenic emissions do contribute to enhanced aerosol

concentrations, but they also cause increased wet deposition, such that higher concentrations of marine biogenic tracers might not be measured at the observatories.

Regarding the potential increase in biomass burning aerosol, there is no observatory indicating a significant positive trend in relevant components (e.g., EBC, absorption coefficient, $nssK^+$). The absence of positive trends might be due to atmospheric transport patterns, meaning observatories might not be located in the regions that are consistently influenced by enhanced biomass burning. Importantly, biomass burning emissions are also ejected higher aloft into the atmosphere, meaning surface stations might not capture these layers due to down-limited mixing in the stratified polar atmosphere. It could also partly be owing to the lack of observations of more specific tracers for biomass burning.

Less snow cover in summer might lead to the hypothesis that enhanced local soil dust emissions can contribute to the aerosol burden at Arctic sites. While soil dust tracer data were available for six stations, none of them showed a significant increase for the overall period, neither during winter nor summer. In contrast, summer $nssCa^{2+}$ declined at Alert, indicating that potentially long-range transport is the more important factor at this station. Moreover, the negative summer trends at Kevo and Pallas, and negative winter trend at Kevo, might reflect decline in anthropogenic contributions rather than natural emissions.

The current absence of significant trends in natural aerosol components might be a result of the choice of aerosol tracers measured at the observatories. While MSA is a unique tracer for marine biogenic emissions, and the $Na^+$ signal is governed by sea salt (Udisti et al., 2016), there is a lack of unique tracers for dust and terrestrial biogenic emissions. Those need to be identified using statistical tools or models. For the latter, observations and speciation of organic aerosol with spatial coverage in the Arctic, and in particular secondary organic aerosol, is desperately needed for better source apportionment (Petäjä et al., 2020; Moschos et al., 2022b). Analyses based on short-term efforts clearly show the potential of various analyses techniques (mass spectrometry or Fourier transform infrared spectral analysis) to isolate signatures of anthropogenic versus natural organic aerosol contribution (Fu et al., 2009; Kawamura et al., 2010; Leaitch et al., 2018; Willis et al., 2017). Isotopic analyses of carbon, nitrogen and sulfur can also provide more details on sources. This has been demonstrated by carbon isotope analyses on atmospheric and deposited aerosols (Rodríguez et al., 2020; Winiger et al., 2019). Nitrogen isotopic analysis has been used to elucidate the $NO_x$ budget in the Arctic, however the study was more focused on trace gas

chemistry rather than aerosol source apportionment (Morin et al., 2008). Sulfur isotopic analysis has been applied a limited number of times for shorter term data sets in the Arctic to distinguish anthropogenic, biogenic and sea salt contributions (Ghahremaninezhad et al., 2016; Seguin et al., 2014). This can also be achieved by online mass spectrometry as has been done at Villum (Nielsen et al., 2019). Also, microphysical properties such as particle size distributions can reveal changes that bulk aerosol

composition cannot. For example, chemical analyses are mostly mass based, and hence changes in particles with smaller diameters are not well captured by such analysis. Those however have already been shown to reflect enhanced contribution of marine biogenic emissions to new particle formation and growth (Dall´Osto et al., 2017; Beck et al., 2020). In addition, optical properties can reflect changes in particle size distributions, for example the SAE at Zeppelin, where the aerosol population seems to evolve

in a way that larger particles dominate. While this is attributed to more frequent observations of sea spray from the Atlantic (Heslin-Rees et al., 2020), the chemical measurements of $Na^+$ at Zeppelin did not capture this change at the $95^{th}$ % significance level. However, a combination of chemical and optical measurements can shed more light on particle origin and driving factors of climate-relevant optical aerosol properties (Quinn et al., 2002). Despite the large data set discussed here, the chemical and optical

observations only coincide marginally. Hence, long-term trends of their co-evolution are not investigated, but such a study would provide very useful insights, particularly adding detailed size distribution information.

**5 Conclusions and outlook**

We presented and analyzed more than a cumulative 1200 years of measurements of aerosol chemical and

895 optical properties recorded at ten surface observatories across the Arctic between 1965 and 2020. Variables include EBC, $SO_4^{2+}$, $NO_3^-$, $NH4^+$, MSA, $Na^+$, $nssCa^{2+}$, $nssK^+$, Fe, scattering and absorption coefficients, *SSA* and *SAE* (see Table 1 and Figure 3).

Annual cycles of all variables clearly show that the wintertime is still dominated by anthropogenic haze across the whole Arctic, while the summer is more strongly influenced by natural aerosol components.

Significantly decreasing long-term trends of anthropogenic tracers provide evidence of the decline of anthropogenic emissions further south which is observed during the haze season at Arctic sites. The

strongest decline in $SO_2$ and BC emissions took place in the 1990s and is visible in large decreases of particulate sulfate and EBC in the North American and European Arctic. Available data from the Eastern Arctic covers only the most recent decades and hence could not capture the change. In the recent past, EBC concentrations seemed to stagnate or slightly increase (although not significantly at the 95[th] percentile significance level). We calculated that concentration increases of at least 5 % per year are currently needed to detect a significant trend per decade, due to the interannual variability of EBC concentrations. Lower rates of change are detectable only over longer periods of time. Future data will show, whether recent efforts by the Arctic Council members in reducing emissions, particularly of black carbon, are effective (Arctic Council, 2017).

Among all long-term trends, 30 % are significant in the haze season where 20 out of 23 show a decline. The three positive trends include $NO_3^-$ at Alert, and scattering coefficient as well as *SSA* at Zeppelin. Despite $NO_x$ emissions decrease at mid-latitudes, particulate nitrate might increase at Alert due to atmospheric chemical processes favoring nitrate partitioning into particles due to the less acidic aerosol (less sulfate). At Zeppelin the enhanced scattering coefficient comes along with a decrease in the *SAE*, indicating a stronger contribution of larger particles, i.e. sea spray from the mid-latitude Atlantic (Heslin-Rees et al., 2020).

For the summer period, no uniform picture of trends has emerged. Twenty-six percent of trends, i.e. 19 out of 73, are significant, whereby five are positive and 14 negative. Negative trends include not only anthropogenic tracers such as EBC at Kevo, but also natural indicators such as MSA and $nssCa^{2+}$ at Alert. Positive trends are observed for $NO_3^-$ at Alert, and $SO_4^{2+}$ at Gruvebadet. The latter could be related to enhanced marine biogenic emissions, but analogous significant trends in MSA were not found there. Only at Kevo the long-term trend is significant and positive. Other tracers, such as $nssK^+$ and $Na^+$, which potentially carry natural signals of biomass burning and sea salt, respectively, only show significant trends at Kevo and Pallas, where they are more likely representative of anthropogenic activities. Hence, no clear evidence of a significant change in the natural aerosol contribution can be seen yet, despite the evident sea ice decline, increased DMS, and increased fire and terrestrial vegetation emissions (Arneth et al., 2016; Galí et al., 2019; Arctic Sea Ice Minimum, 2020; Rogers et al., 2020). This points towards five main action items for monitoring Arctic aerosol.

-    First, long-term observations targeted at natural aerosol tracers are needed. Increased natural particle and precursor emissions might produce signal enhancements, which are small compared to the inherent natural signal variability. Hence, new trends might not yet be visible but might emerge in the years to come. They are important indicators of feedback mechanisms in Arctic amplification.

-    Second, the location of an observatory matters for which trends might emerge as significant or not. Not every observatory will exhibit all types of natural aerosol changes clearly. For example, Becagli et al. (2019) find a difference in MSA trends between Gruvebadet and Thule, likely due to different evolution of marine ecosystems. A natural change hot spot map of the Arctic combined with simulations of atmospheric transport and aerosol processes can suggest areas that require
new measurement efforts. Moreover, most stations are located near sea level, and only Summit provides valuable insights into processes further aloft including in the free troposphere. Only comprehensive measurements at all Arctic observatories discussed here, and ideally additional ones in the Eastern Arctic, are able to address the complexity and variability of aerosol sources and processes across the region.

-    Third, an assessment of whether the currently monitored aerosol properties are appropriate to capture changes in natural aerosol compounds is needed. Source apportionment based on speciated organic aerosol components is currently lacking except for a few short-term case studies (Moschos et al., 2022b). Those together with isotopic analyses of carbon, nitrogen and sulfur can shed light into natural aerosol changes. Microphysical properties such as the size distribution and
inherent optical characteristics such as the single scattering albedo and scattering Ångström exponent can further reveal changes in the number, size and composition and thus origin of particles, which chemical analyses alone, especially of bulk aerosols, cannot. Moreover, aerosol microphysical and optical properties are directly connected to aerosol climate effects.

       -    Fourth, continuing the existing long-term efforts is essential. Only long-term time series can
document Arctic change and further our understanding of Arctic climate and biogeochemical processes.

-   Fifth, standard operation procedures for the measurement of particle properties are needed such that they can be compared (e.g. same cut-off size on the inlets). For optical properties the Global Atmosphere Watch network provides recommendations for harmonized measurements. Furthermore, observatory operators should be encouraged to publish the data timely in established formats (see details in Laj et al. (2020)).

In summary, aerosol properties are an important tool to analyze changes in anthropogenic and natural atmospheric chemical influences on the Arctic. Tremendous monitoring efforts have already been conducted, which have helped to understand the implications for past, present and future Arctic climate change. The decision to start regular monitoring of key anthropogenic components in the early 1990's - or even before – turned out to be important and timely, because it enabled us to capture a remarkable change in the concentrations of anthropogenic climate-forcing agents in the Arctic lower atmosphere that occurred in the late twentieth century. Establishing specifically targeted observations for the ongoing and future change of natural aerosols is essential to developing improved scenarios of their climatic and biogeochemical effects on the Arctic. Continuation of the monitoring programs targeting changes in variables during the haze season linked to anthropogenic emissions at mid-latitudes is a prerequisite to give further evidence to the declining trends determined today and for their quantification feeding global climate and earth system models.

**Acknowledgements**

We thank the entire Arctic Monitoring and Assessment Programme Expert Group on Short-lived Climate Forcers for inspiring conversations and feedback. For Alert, authors would like to thank operators, students and technicians for the aerosol program operations, calibrations and maintenance of instruments, especially Dan Veber for maintenance of optical and physical measurements instruments and Alina Chivulescu for aerosol sample chemical analysis. CFS Alert for maintenance of Alert base station. The Villum Foundation is gratefully acknowledged for financing the establishment of the Villum Research Station. Bjarne Jensen, Christel Christoffersen, and Keld Mortensen from Aarhus University is gratefully acknowledged for their technical support. Thanks to the Royal Danish Air Force and the Arctic Command for providing logistic support to the project. The data from Gruvebadet were achieved thanks to Projects PRIN- 20092C7KRC001 and RIS 3693 "Gruvebadet Atmospheric Laboratory Project (GRUVELAB)" and to the coordination activity of National Council of Research (CNR), which manages the Italian Arctic Station "Dirigibile Italia" through the Institute of Polar Sciences (ISP). For Barrow / Utqiagvik, PMEL would like to thank all of the station operators who have made sample collection possible over the past years. This is PMEL contribution number 5287. For Barrow / Utqiagvik, Summit and Tiksi the authors would like to thank the operators for the care and feeding of chemical and optical instruments and Derek Hageman for his continued excellence in data acquisition, processing and archiving of data. The Arctic and Antarctic Research Institute (AARI) and the Finnish Meteorological Institute (FMI) are acknowledged for Tiksi measurements and data. FMI is acknowledged for Pallas measurements and data. We thank all who have carried out long-term measurements at Arctic observatories, and made the data free available through EBAS. The authors also thank two anonymous referees for their highly valuable comments.

**Financial support.**

This work received support from the Swiss National Science Foundation (grant no. 200021_188478). J.S. holds the Ingvar Kamprad Chair for Extreme Environments Research sponsored by Ferring

Pharmaceuticals. This research has been financially supported by the Danish Environmental Protection Agency and the Danish Energy Agency, with means from MIKA/DANCEA funds for environmental support to the Arctic region (project nos. Danish EPA: MST-113-00-140; Ministry of Climate, Energy, and Utilities: 2018-3767) and ERA-PLANET (The European Network for observing our changing Planet) projects, as well as by iGOSP, iCUPE. Measurements and data evaluation for aerosol optical properties at Barrow / Utqiagvik are supported by DOE/ARM (ANL award# 0F-60239), measurements at Summit occur under the aegis of NSF and NOAA, and aethalometer measurements at Tiksi occurred through NOAA cooperation with Roshydromet that ended in 2018. Pallas and Tiksi aerosol measurements are supported by Academy of Finland Flagship funding (grant no. 337552) and by Horizon 2020 ACTRIS IMP project (grant agreement 871115).

**Code/Data availability**

Most data are publicly available on http://ebas-data.nilu.no/ or else on request via the personal communication contacts listed in Table 1. All these contacts are co-authors and in charge of observatory data, which needs to undergo specific control before it can be posted publicly. This publication partly precedes open access data publication. All data will be available via http://ebas-data.nilu.no/ in the future.

**Author contributions**

LU, PKQ, EA, AM, JBP, SS, SD, KE, PH, JL, JP processed and analyzed data from the observatories. JS, SS, and SD calculated annual cycles and seasonal trends. KVS provided emissions data. JS wrote the manuscript. All authors contributed to interpretation and commented on the manuscript.

**Competing interests**

The authors declare no competing interests.

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
