# Peer review of "Pan-Arctic seasonal cycles and long-term trends of aerosol properties from ten observatories"

_Atmospheric Chemistry and Physics, 2021_

## Author Comment (AC1)

**Reply to reviews**

RC = Reviewer Comment

AR = Author Reply

**General remarks to the editor and both reviewers.**

Please note that the addition of new datasets warrants new co-authorships. We include now: Stefania Gilardoni, Mauro Mazzola, James Laing, and Philip Hopke. See revised author list.

The addition of new data also means that we have added specific text. Please see the revised manuscript in tracked changes. More specifically we added discussion on the Pallas and Kevo chemical data, and Gruvebadet optical aerosol properties. The overall number of significant trends has changed, and we now provide an overview in a new table 3.

The acknowledgements have been updated.

We now refer to the NOAA Barrow observatory as Barrow / Utqiagvik, because the town has been renamed recently and both names appear in the literature now.

**Reviewer 1**

**RC1.1:** Schmale e et al. present a comprehensive and well written overview of the aerosol properties and trends at the Arctic sites over several decades. Arctic is changing rapidly, and this study is a good benchmark for further assessments of potential changes in aerosol properties in future climate and emission regimes.

I think the study can be published with some improvements suggested more specifically below. I would especially encourage the authors to make use of the 30 years of SIA observation (especially $SO_4^{2-}$, $Na^+$, $Ca^{2+}$) at Zeppelin and not only use a dataset with much shorter time period. Pallas also have observations of SO4 since 1996 and $Na^+$,$Ca^{2+}$,$K^+$ since 2003, which the paper would benefit from using. All these data are available in EBAS.

**AC1.1:** We thank the reviewer for the positive feedback and assessment, as well as the very helpful comments. We agree that the full time series at Zeppelin as well as at Pallas should be used. In fact, we had a nice response in the scientific community with the publication of our preprint and have additional data sets. We now include the following additional or prolonged datasets:

| Station | Variable | Period | Change compared to original manuscript |
|---|---|---|---|
| Zeppelin | $SO_4^{2-}$, $Na^+$, $Ca^{2+}$, (note there is a known issue with $NO_3^-$ and $K^+$ for the past years, hence we included data up to 2009 only) | 1993 - 2021 | Longer time period |
| Pallas | $SO_4^{2-}$, $NH_4^+$, $Ca^{2+}$, $NO_3^-$, $Na^+$, $K^+$ | ~2003 – 2020 | New data |
| Kevo | $Na^+$, $K^+$, $Ca^{2+}$, MSA, $SO_4^{2-}$ | 1964 – 2010 | New data |
| Gruvebadet | scattering and absorption coefficient | 2010 - 2020 | New data |
| Barrow / Utqiagvik | Scattering coefficient | 1978 - 2020 | Longer time series |

More specific comments:

**RC1.2:** Line 90. Maybe add references and numbers from the IPCC report recently published, which gives the must updated assessment of the impact of SLCF

**AR1.2:** We added the following line below, based on the newest result from AMAP that came out in the meantime. AMAP is more targeted towards the Arctic than the IPCC AR6, hence we refer to these results. Added text:

*The summary for policy makers of the most recent AMAP report shows that the Arctic has warmed by 0.28°C per decade between 1990 and 2015 due to reductions in $SO_2$*

*emissions. Reduction in BC emissions have led to a cooling of about 0.06°C per decade, whereas the CO₂ increase contributed 0.29°C per decade* (AMAP, 2021).

- **RC1.3:** Line 96. Before the establishment of the Zeppelin observatory monitoring was conducted in Ny Ålesund. Thus observations at Svalbard foes back to the seventies.

  **AR1.3**: Thank you for pointing this out. We have corrected this in the manuscript.

- **RC1.4:** Line 160: The results from Collaud Coen et al (2020) should be added here.

  Good point. We have added the following:

  *The aerosol absorption coefficient can also be used to describe the influence of black carbon, an absorbing aerosol. Collaud Coen et al.* (2020b) *performed trend analysis on annual data and found significant trends at Barrow / Utqiagvik over a 10 year horizon of -9.91 % per year and -0.90 % per year at Tiksi. Alert features a positive trend of 2.77 % per year over 10 year horizons. These data are until 2016-2018.*

- **RC1.5:** Line 166. Do not really see it is a contrast to Platt et al 2021. Different time periods and season contra annual trends. Maybe use another word than contrast.

  **AR1.5:** We rephrased to: *For Zeppelin, Platt et al (2021) find a decline of 44 % between 1990 and 2019. The difference to findings by Hirdman et al. (2010a) can be explained by the use of different data sets and seasonal vs annual aggregates.*

- **RC1.6:** Line 182. Not sure if I understand this statement correctly. "no increased contribution during the fire season has been shown so far." Are you talking about trend studies or influence by fires. Surely there are several examples of high episodes observed at the Arctic sites caused by emissions from fires (e.g. Stohl et al 2007)

  **AR1.6:** We have clarified the statement: *"... no increase in the trends during the fire seasons has been shown so far."* We only refer to the trends, not individual observations, which clearly demonstrate impact as the reviewer points out.

- **RC1.7:** Line 327. Not clear how you have selected valid time series for trend analysis. With "at least five consecutive years" does that mean you can make a decadal trend with only five years? You need at least 7 years of data to use the Mann Kendall and Theil-Sen slope method.

  **AR1.7**: We originally computed trends if there were at least 5 consecutive years of data. Collaud Cohen et al. (2020a) show that computing trends > 4 years is possible, while not advisable. Following the reviewer's comment, we have now eliminated all indications of decadal trends, if the number of years per decade is smaller than 8. This criterion is stricter than suggested by the reviewer, because there are several decades where an eight-year coverage is available.

- **RC1.8:** Line 353 contradicts a bit with what is written in line 357 (EBC vs SO4 seasonal trends respectively at Zeppelin compared to other sites). Sulfate is also easily washed out by rain and one would maybe expect similar behaviour. The differences may be due to different time periods used for comparing the seasonality between sites.

**AR1.8**: Thank you for pointing this out. This makes sense. We have replaced the sentence starting "This is likely due..." with *"The difference between the stations might be due to the comparison of different years and not necessarily physical aerosol processes."*

- **RC1.9:** Line 526. Even though the economic breakdown in Eastern Europe (and Soviet Union) results in a relatively steep decline in SO2 emissions in the nineties, the main reason for the large reductions in Europe are due to international protocols agreed upon under the UNECE LRTAP and EU directives, and in North America by the Clean Air Act.

AR1.9: That is correct. To clarify this we modified the respective sentence to*: "SO$_2$ emissions, a precursor to particulate SO$_4{}^{2-}$, decreased strongly in Europe and the Western Arctic countries since 1980 due to clean air policies such as UNECE LRTAP and the US Clean Air Act. SO$_2$ emissions showed a clear dip between 1990 and 2000 in the Eastern Arctic, probably due to the economic change in the former Soviet Union."*

- **RC1.10:** The increase in So4 during summer is not seen in the regular monitoring at Zeppelin for the period 1990-2019 nor for shorter time periods. See figure

**AR1.10**: No reference line is given for this comment, but we suppose it is the original l. 557 *"... for Zeppelin and Gruvebaded (...), where SO$_4{}^{2-}$ trends are positive and significant during summer."* With the updated Zeppelin dataset, it is true that there is no significant trend. We have added to the sentence "..., where only for Gruvebadet SO$_4{}^{2-}$ trends are...".

- **RC1.11:** The dust tracer (Ca and Fe) measured are probably mainly water soluble (at least Ca2+) and may not be a good indicator of the real influence of dust.

**AR1.11:** This is an important point, that we only consider soluble species. With the hypothesis that calcium originates either from sea salt or dust, we represent the soluble fraction of calcium in dust. In how far this fraction is a good proxy for the "real influence of dust" is difficult to say in the absence of other relevant measurements. To account for this, we now write in l. 701: *"NssCa$^{2+}$, a **soluble** tracer for mineral and soil dust, shows a negative trend at Alert in summer, which is consistent with the results by Sharma et al. (2019). "*

- **RC1.12:** Surly using harmonized standard methods for aerosol optical properties and other aerosol observations are extremely important, and this is being developed. The AMAP (Arctic) sites should follow recommendations made by WMO/GAW. In addition, the observations should be made available and reported in a standardized protocol to international frameworks (GAW, AMAP, EMEP etc. See i.e. discussion by Laj et al 2020 :

**AR1.12:** Thank you for this input. We added in the conclusions: *"For optical properties the Global Atmosphere Watch network provides recommendations for harmonized measurements. Furthermore, observatory operators should be encouraged to publish the data timely in established formats. See details in Laj et al. (2020)."*

- **RC1.13:** Figure 1 in *: spelling error. Sind should be since. Find the reasoning for why not the longest time series have not been chosen quite strange. Why not use the longest one available.

   **AR1.13:** We have replaced "sind" with "since".

- **RC1.14:** Figure 2, in fig h) should that be red and not orange? Not Zeppelin data displayed

   **AR1.14:** Thank you for spotting this. The trace is now red, pointing to Alert.

Table 1:

- **RC1.15:** It would have been useful to add for which time periods you are using data from for all the species

   **AR1.15:** This information is contained in Figure 3 (former Table 2), see grey areas. Given that Table 1 appears earlier, we now also include this information in Table 1.

- **RC1.16:** Seems a bit strange to distinguish EBC from absorption coefficient since these are derived from the same instrument.

   AR1.16: While this might seem strange for readers who are very familiar with the conversion, it might not be evident for readers who have a different background. We believe that this manuscript is interesting for a large audience, including the climate modeling community, and hence decided to keep the EBC values, because EBC has received a lot of attention in the Arctic climate change context. Furthermore, the conversion allows to compare to EC measurements, which are done at several stations. We indicate now in Table 2 all cases where EBC was derived from the absorption measurements, how that was done (MAC values and $C_{ref}$ if applicable).

- **RC1.17:** Alert: Why are Absorption and scattering data taken from personal communication and not from EBAS. 2005-2019 available and also used by Collaud Coen et al. in their trend study

   **RC1.17:** Thank you for pointing out this inconsistency. We have corrected it to "EBAS".

- **RC1.18:** Zeppelin: As mentioned above, I find it strange that you have not used the SO4,Na,Ca, from EBAS (and SO4 presented in Platt et al 2021), which has much longer time series than what is available by Sharma et al (2012). Further in Sharma et al, only MSA data is presented even though they probably collected many other species. The EBC data (or absorption rather) and scattering data are also available in EBAS as presented in Collaud Coen et al.

**AR1.18:** We have now included the much longer time series from EBAS, see AR.1.1 and refer in Table 1 also to EBAS.

Reviewer 2:

Review of "Pan-Arctic seasonal cycles and long-term trends of aerosol properties from ten observatories" by Schmale et al. in ACPD.

**RC2.1:** The manuscript presents a coherent analysis of an extensive Arctic dataset on aerosols and aerosol properties, focusing on uncovering their seasonal and decadal variability. Both natural and anthropogenic aerosol tracers are considered in the analysis and manuscript discussion, accurately highlighting the complexity of the issue. The manuscript describes nicely how different sources are important on different sides and at different times showing that the Arctic is not a homogenous area in this respect. The manuscript is well written, enjoyable to read and makes an important contribution for the scientific field.

Most of the data for the analysis are gathered from a publicly available nilu/ebas database. Although a fraction of data with supporting conclusions have been presented in previous literature, the systematic and comprehensive analysis and the new, previously unpublish data bring clearly new perspectives and conclusions. I recommend publishing the manuscript in ACP after modifications. I consider that the requested modifications do not require a major work and are therefore minor.

**AR2.1:** We thank the reviewer for the positive and constructive feedback.

General comments:

**RC2.2:** My main general comment is about the description of the methodology. The manuscript concludes (action item 4) on the importance of standardized measurement and data practices. Was this met here?

   a) For example, was ebas level2 data always used? If so, were the data that were taken from personal communication treated (and measured) similarly? This should be mentioned in the text.
   b) Why Tiksi nephelometer was measuring at "ambient humidity"? How much was the RH?
   c) How the data flags (in ebas) were considered in data analysis? Which flags were included and which were omitted from final analysis?
   d) Would you please consider adding some relevant information on at least these, and other according to your consideration, in the methods section and data availability section (also see my comment regarding Table 1). In principle, information should be sufficient to repeat this study.

**AR2.2:** Thank you very much for this comment. The precise questions are very helpful in providing the necessary information to make the study reproducible.

   a) Yes. Ebas level 2 was always used. Please see the updated Table 1 now, where several of the "personal communication" datasets were replaced by EBAS data. In these cases, we obtained the EBAS data from the indicated person, but the dataset is the same in EBAS. To avoid confusion, we have harmonized that in the Table. With regards to the remaining "personal communication" data, most of them were/are in preparation for

EBAS and are hence of similar data quality and were also treated comparably to actual EBAS data. This concerns Villum, Gruvebadet, and Kevo. The Barrow / Utqiagvik EBC data are from EBAS absorption data and were converted according to the newly inserted Table 2.

b) The Tiksi scattering data is from EBAS and the file header provides the following information: *Inlet description: PM10 at ambient humidity inlet, flow 16.67 l/min; Humidity/temperature control: None; Humidity/temperature control description: passive, sample heated from atmospheric to lab temperature*. There is not RH data indicated in the file, so we cannot comment on this.

c) Data flags: We only discarded data if the data flags are marked red in the EBAS data submission manual for the respective aerosol property, https://ebas-submit.nilu.no/temps.

d) In section 2.1, we added: *"In this study, we only used quality checked and assured datasets from the sources provided in Table 1. EBAS data are level 2, and only data points with a red flag code as given in the EBAS data submission manual (https://ebas-submit.nilu.no/temps) were omitted. Data from other sources or personal communication are equivalent to EBAS standards."*

We also added for the Tiksi Nephelometer entry in Table 1: *Note, measurements at ambient humidity when > 40 % might result in larger values compared to RH< 40 % values."*

**RC2.3:** Another minor general comment I have on the discussion of primary aerosol sources (mainly sea spray and dust here). Their concentrations are not solely dependent on the changes in air flow routes and availability of the source (open sea water, open land areas) but the resuspension depends largely on the winds (speed). The seasonality of the winds over different sea areas around the Arctic could be mentioned somewhere in the text, and maybe some information added if there were notable changes in storms or other wind patterns in recent decades?

**AR2.3:** This is an important consideration. In l. 263 (original counting) we have added:

*"Generally, the Arctic is windier in winter than in summer, with the highest wind speeds between January and April, and the lowest in July and August, based on the 40 year ERA5 climatology (Rinke et al., 2021). Based on a trend analysis on several re-analysis products Vessey et al. (2020) found that there is no significant change in storminess in any season in the Arctic. This does however not mean that there can be Arctic regional changes over time, which are not captured by Arctic wide data aggregates (Atkinson, 2005)."*

**RC2.4:** As a final general comment I add that from northern Scandinavia (e.g. Pallas) there are additional long-term data series on aerosol chemical composition available in ebas that the authors could consider adding in the analysis if they find that those could add to the conclusions.

**AR2.4:** Thank you very much for pointing this out. In fact, we have been made aware of additional datasets by the community response to this preprint. We now include the following additional or prolonged datasets (see also AR1.1 to reviewer 1):

| Station | Variable | Period | Change compared to original manuscript |
|---|---|---|---|

| Zeppelin | $SO_4^{2-}$, $Na^+$, $Ca^{2+}$, (note there is a known issue with $NO_3^-$ and $K^+$ for the past years, hence we included data up to 2009 only) | 1993 - 2021 | Longer time period |
|---|---|---|---|
| Pallas | $SO_4^{2-}$, $NH_4^+$, $Ca^{2+}$, $NO_3^-$, $Na^+$, $K^+$ | 2003 – 2020 | New data |
| Kevo | $Na^+$, $K^+$, $Ca^{2+}$, MSA, $SO_4^{2-}$ | 1964 – 2010 | New data |
| Gruvebadet | scattering and absorption coefficient | 2010 - 2020 | New data |
| Barrow / Utqiagvik | Scattering coefficient | 1978 - 2020 | Longer time series |

Minor comments:

**RC2.5:** p3. l100. Although aerosol optical parameters in Pallas have been measured since 2000, aerosol number and ionic composition measurements in Pallas started earlier, in 1990s (data in ebas). Since this sentence refers to Arctic pollution monitoring maybe a better reference of the historical development of Pallas research activities and monitoring would be Lohila et al., 2015:

Lohila A., Penttilä T., Jortikka S., Aalto T., Anttila P., Asmi E., Aurela M., Hatakka J., Hellén H., Henttonen H., Hänninen P., Kilkki J., Kyllönen K., Laurila T., Lepistö A., Lihavainen H., Makkonen U., Paatero J., Rask M., Sutinen R., Tuovinen J.-P., Vuorenmaa J. & Viisanen Y. 2015: Preface to the special issue on integrated research of atmosphere, ecosystems and environment at Pallas. Boreal Env. Res. 20: 431–454. http://hdl.handle.net/10138/228278

**AR2.5:** We have added the reference.

**RC2.6:** p4. l119. Consider starting "the surface aerosol observations", since it is not clear what "observations" are meant, especially after a long paragraph solely on eBC.

**AR2.6:** Done.

**RC2.7:** p5. l147. Tiksi (not Tiski)

**AR2.7:** Done.

**RC2.8 :** p5. l154. Consider re-phrasing. It is not clear what is "the same database" used for "the study" of annual cycles. Assuming you mean: https://doi.org/10.21336/gen.1, would be clearer to write that directly.

**AR2.8:** We rephrased: *"The same data as discussed above in the context of annual cycle studies have been used for the purpose of long-term trend analyses."*

**RC2.9:** p6. l183. "This is likely due to the fact that wildfires emit BC and OC further aloft into the atmosphere and in the absence of down-mixing to the boundary layer such aerosol layers are mostly not captured by surface observations." Any reference to support this statement? Indeed, the fire emissions vertical profile vs. transport dynamics is an important (open) question, for which it is a quite strong statement to say this is known as a fact (e.g. Remy et al., 2017, ACP or Ke et al., 2021, JGR).

**AR2.9:** We have rephrased to: *"This could be related to the wild fire plume injections heights, which might reach higher altitudes and BC and OC could then be transported further aloft. In the absence of...not captured by surface observations. However, knowledge on plume injections heights are still uncertain (Rémy et al., 2017; Ke et al., 2021)."*

**RC2.10:** p7. l198-199: I would suggest to add "summertime nss-SO4", because the trend studies, both winter and annual averages, do exist and have also been comprehensively summarized in this work which makes this sentence slightly confusing. In general, nss-SO4 is a complicated "tracer" since its sources are both natural and anthropogenic. This contradictory is also reflected in the article introduction indirectly.

**AR2.10:** We have added "summertime" to clarify.

**RC2.11:** p10. l268. Figure 1 does not show data.

**AR2.11:** We have exchanged "data" with "locations".

**RC2.12:** p11. l287. Add the detection limit here e.g. in parenthesis.

**AR2.12:** Our formulation was misleading: 0.045 Mm-1 is the detection limit. *"Values below the limit of detection of 0.045 Mm$^{-1}$ ..."*

**RC2.13:** p15. L311. What do you mean with "prior deseasonalization"?

**AR2.13:** For a deseasonalization, or seasonal adjustment, we would have define the seasonal component of the data. This seasonal component has to be removed prior to trend calculation, because the magnitude of the seasonal signal is larger than any trend. To avoid fitting the data for the seasonal component, we simply calculate seasonal trends, hence the seasonal component is the same for each year and cancels out. We clarify: *"... prior deseasonalization, i.e. removal of the seasonal component, of the data."*

**RC2.14:** p17. l362. Could you be more specific with what is meant by "ammonium's statistical distribution"?

**AR2.14:** We replaced "statistical distribution" with *"The box and whisker plots at Villum and Alert are almost identical."*

**RC2.15:** p22. l431. Even though it's difficult to compare the bars in fig2 only, but for Tiksi it looks like the absorption is much higher in comparison to Barrow and Alert (10-fold) than the EBC is (3-fold), and not really 1:1? Check.

**AR2.15**: The EBC value was derived from an Aethalometer without a $C_{ref}$ value. When converting the Aethalometer attenuation data to the absorption coefficient it is however important to apply a $C_{ref}$ value to make the data comparable with filter based absorption measurements. We now applied a $C_{ref}$ value of 3.5, which means that also the absorption data is now different by a factor 3, roughly, only. To make all conversions transparent, we have added a new Table 2.

**RC2.16:** p28 l 504-506. First sentence says that emissions rise in the 2000s. Second sentence says that EBC concentrations do not show decline, likely due to short measurement period and variability. Why is it assumed that a decline should be seen, especially if emissions increase?

**AR2.16:** Thank you for spotting this. We have reformulated to: *"After the year 2000 emissions slightly declined in the Eastern Arctic and rest of Europe. In the Western Arctic they slightly increased towards 2005 and then decreased towards 2008 while then staying at the same level. In Asia emissions were on the rise until 2008 and then declined, while in the rest of the world emissions increased. Arctic EBC concentrations, which are mostly impacted by the Eastern and Western Arctic as well as Europe, do neither show a decline or incline from 2000 onwards during winter."*

**RC2.17:** p29. l529. What is "this" pattern? The pattern observed in Eastern Arctic countries, or elsewhere, or Asia..?

**AR2.17:** We rephrase to: *"... follow the pattern of the strong emission decline in the 1990, showing the strongest decline in this decade and then leveling off."*

**RC2.18:** p32. l631. "They" should be "they"

**AR2.18:** Corrected.

**RC2.19**: p33. l638-641. Unclear sentence, rephrase.

**AR2.19**: We rephrased to: *"However, using linear regression Quinn et al. (2002) have shown for data between 1997 and 2000 that $Na^+$ exerts a dominant role on the winter aerosol scattering coefficient, while $SO_4^{2-}$ dominates the spring scattering."*

**RC2.20:** p34. l663. Interesting that here Pallas shows negative SAE trend in summer while previous studies (Collaud Coen et al 2013; Lihavainen et al., 2015) have shown this trend for winter. But as pointed out several times in the manuscript, the trends are also sensitive to the method and with such short time series should be interpreted with caution.

**AR2.20:** We added in l. 663: *"Collaud Coen et al. (2013) and Lihavainen et al. (2015) showed a significant negative trend for that period in winter."*

**RC2.21:** p41. l850. Might consider adding that the aerosol microphysical properties also provide data on aerosol quantities that are directly connected with their climatic impacts.

**AR2.21:** We have added: *"Moreover, aerosol microphysical and optical properties are directly connected to aerosol climate effects."*

**RC2.22:** p40-41 action items / conclusions: Conclusions and action items very strongly seem to highlight the need for additional monitoring and analysis of natural Arctic aerosols. Why is this emphasized? Or do you mean there is a lack of data on the aerosol organics, more specifically? In my view, the results presented support the need for intensified aerosol monitoring around the Arctic (due to complexity and variability of the sources), importance of long-term efforts (for understanding the trends) and need for interdisciplinary collaboration (for complexity of the sources and processes).

**AR2.22:** This is a good summary of the key points we would like to make. In addition, we find it important that natural aerosol tracers need to receive a stronger focus, because they can tell us about climate-feedback mechanisms. This is our first point. The second point goes into the direction of "complexity and variability of sources". To reinforce this point we added in l. 842: *"Only comprehensive measurements at all Arctic observatories discussed here, and ideally additional ones in the Eastern Arctic, are able to address the complexity and variability of aerosol sources and processes across the region."* Point three touches upon the need to have more targeted observations which go beyond just mass based methods (and optical properties). This goes logically together with point 1, so we have swapped point 2 and 3. The fourth point encourages data sharing and comparability so comprehensive analyses are possible. It is true that the long-term aspect is not mentioned as individual point, so we have added before the original point four a new one saying: *"Fourth, continuing the existing long-term efforts is essential. Only long-term time series can document Arctic change and further our understanding of Arctic climate and biogeochemical processes."*

Tables:

**RC2.23**: Table 1.

a) Sometimes both an article and ebas are indicated as data sources. Why? If an article is provided where the data are used and measurements described more in detail, it should be done systematically for every dataset, or any. Was the article (e.g. in case of Gilardoni, Dutkiewich, Heslin-Rees, etc.) really the data source or a personal communication?
b) Does ebas provide EBC values for absorption measurements (Alert, Tiksi)? Using what MAC?
c) For Alert, Barrow and Tiksi, what measured wavelength was used to convert data to abs coef @550 nm?
d) In addition, would you be able to provide a link to specific datasets in ebas and access date, instead of referring to a general database? I understand if this last point is adding too much data in table, then current form is ok.

**AR2.23:** We agree with the reviewer that the data sources were not particularly clear. We have updated this information in the table.

a) In the case when the data used in a paper are also available via EBAS and have not been treated differently as indicated in AR2.2 (and now in the methods section), we now indicate EBAS as source for consistency. If the reference to the paper remains, we obtained the data from the authors in their final form as used in that publication. We clarify this with *"If a manuscript is given as data source, data were obtained in their published form from the authors."* in the title of the table.
b) Please the new Table 2 for details how data were converted to EBC.
c) We have indicated this for all stations in the new Table 2 now.
d) While we agree that it would be very convenient for the readers to have the direct links, we also think that searching EBAS is can be quickly done by the readers themselves and they can then access the time period and variables of their choice directly.

Figures:

**RC2.24:** Figure 1. Missing Pallas station.

**AC2.24:** added.

**RC2.25**: Figure 2. Tiksi EBC concentrations (1$^{st}$ panel) appears to be about 3 times higher than in Alert and Barrow, however, the absorption is 10-fold? Is also Tiksi scattering multiplied by 0.1? Color shading for grey and black are not possible to separate from each others.

**AR2.25**: For Tiksi absorption and EBC please see AR2.15. The scattering is indeed multiplied by 0.1 as well. Thank you for spotting that we did not indicate this. We have changed the color for Summit to pink.

**RC2.26:** Figure 4 (and corresponding text). Is western Arctic including all the continental territory of USA and Canada? And does Eastern Arctic include all the Russia? Is that

overlapping with Asia? Reference to AMAP, 2021 is given, but this report does not have a reference yet so a short explanation here could be given. In addition, should $NO_2$ be $NO_x$?

**AR2.26:** $NO_2$ is indeed $NO_2$ and not $NO_x$. The regional definitions are: Western Arctic Council (Canada and United States, **entire countries**), and Eastern Arctic Council (Kingdom of Denmark, Finland, Iceland, Norway, the Russian Federation, and Sweden, **entire countries**). We have added this to the caption of Fig. 4.